# Learning Invariant Causal Mechanism from Vision-Language Models

**Zeen Song** [* 1 2]  **Siyu Zhao** [* 2]  **Xingyu Zhang** [* 1 2]  **Jiangmeng Li** [1]  **Changwen Zheng** [1]  **Wenwen Qiang** [1]

## Abstract

Contrastive Language-Image Pretraining (CLIP) has achieved remarkable success, but its performance can degrade when fine-tuned in out-of-distribution (OOD) scenarios. We model the prediction process using a Structural Causal Model (SCM) and show that the causal mechanism involving both invariant and variant factors in training environments differs from that in test environments. In contrast, the causal mechanism with solely invariant factors remains consistent across environments. We theoretically prove the existence of a linear mapping from CLIP embeddings to invariant factors, which can be estimated using interventional data. Additionally, we provide a condition to guarantee low OOD risk of the invariant predictor. Based on these insights, we propose the Invariant Causal Mechanism of CLIP (CLIP-ICM) framework. CLIP-ICM involves collecting interventional data, estimating a linear projection matrix, and making predictions within the invariant subspace. Experiments on several OOD datasets show that CLIP-ICM significantly improves the performance of CLIP. Our method offers a simple but powerful enhancement, boosting the reliability of CLIP in real-world applications. The source code is available at https://github.com/ZeenSong/CLIP-ICM.

## 1. Introduction

As one of the most successful large-scale pre-trained vision-language models, Contrastive Language-Image Pretraining (CLIP) (Radford et al., 2021) has garnered significant attention. Trained on 400 million diverse image-text pairs, CLIP achieves robust cross-modal alignment without task-specific supervision. This enables direct generalization to unseen

tasks by measuring semantic similarity between images and textual category descriptions (Radford et al., 2021). Notably, CLIP performs well in out-of-distribution (OOD) scenarios, effectively adapting to domain variations and recognizing previously unseen categories.

However, this ability of CLIP is task-independent. In many real-world applications, CLIP requires fine-tuning to adapt to specific tasks (Gao et al., 2023; Zhou et al., 2022a; Shu et al., 2023; Zhang et al., 2023b). When fine-tuning CLIP for downstream tasks, a critical question arises: Can CLIP maintain its strong capabilities when faced with test data that differs from the training distribution? More importantly, can it still perform well when encountering new classes not seen during training? To address this, we conduct an experiment on the Terra Incognita dataset (Beery et al., 2018), which contains species data collected from diverse real-world environments. The results in Table 1 show that fine-tuning CLIP in the training domain does not necessarily lead to good performance in the test domain. Furthermore, when encountering new classes not observed in the training domain, fine-tuning on the training domain can harm the zero-shot ability of CLIP.

To address the above problem, we find that inconsistent prediction issues can be well explained from a causal perspective. Inspired by (Pearl, 2009; Pearl & Bareinboim, 2014; Schölkopf et al., 2021), we first propose a Structural Causal Model (SCM) to capture the causal processes underlying OOD predictions of CLIP. In this causal model, predictions rely on two types of latent factors, which are assumed to generate the image data: (1) environment-invariant factors that remain unchanged across different environments and (2) environment-variant factors that vary with the environment. By analyzing the causal mechanisms in the SCM, we observe that if predictions rely on both invariant and variant factors, the causal mechanism in the training environment differs from that in the test environment. However, when based solely on invariant factors, the prediction mechanism remains consistent across environments. This raises an intriguing question: could we design a predictor that relies only on the invariant factors?

A key challenge in obtaining such an invariant predictor is that it is unclear which parts of the extracted representations of CLIP correspond to the invariant factors (Hyvärinen &

---

[*]Equal contribution  [1]Institute of Software Chinese Academy of Sciences, Beijing, China [2]University of the Chinese Academy of Sciences. Correspondence to: Wenwen Qiang <qiangwenwen@iscas.ac.cn>.

*Proceedings of the 42nd International Conference on Machine Learning*, Vancouver, Canada. PMLR 267, 2025. Copyright 2025 by the author(s).

Pajunen, 1999; Locatello et al., 2019). By analyzing the identifiability of CLIP, we first demonstrate that the representations learned by CLIP can be viewed as a linear combination of invariant and variant factors. Under this condition, we prove that, if interventional data (Refer to Appendix H.1 and Section 6 for details) are available, it is possible to derive a linear projection matrix that maps the representations of CLIP to the invariant factors. The learned projection matrix maps the image and text representations of CLIP into the same invariant space. In this space, it becomes possible to perform invariant prediction. We further analyze the OOD risk of this invariant predictor compared to a regular predictor. We provide conditions where the invariant predictor indeed achieve lower OOD risk.

Guided by the above theoretical analysis, we propose a novel modeling framework called the *Invariant Causal Mechanism of CLIP* (CLIP-ICM). This framework consists of three stages: (1) Collect the interventional data. (2) Estimate a linear projection to invariant subspace. (3) Perform invariant prediction in the invariant subspace. Under the CLIP-ICM framework, we propose specific methods to obtain interventional data using either image or text data. Additionally, we propose a learning objective for estimating the projection matrix. Notably, our method does not require retraining the backbone network of CLIP. We evaluate the proposed CLIP-ICM on OOD generalization datasets, including Domainbed (Gulrajani & Lopez-Paz) and variants of ImageNet (Recht et al., 2019; Hendrycks et al., 2021b;a; Wang et al., 2019). Experimental results demonstrate the outstanding effectiveness of our approach. Considering its low computational cost and significant performance improvement, our method is both simple and effective.

Our contributions can be summarized as follows: 1) We identify and demonstrate that CLIP exhibits inconsistent performance in the train and test domain when fine-tuned in OOD generalization scenarios. Through an in-depth causal analysis, we demonstrate that it is possible to achieve consistent predictions using only invariant factors. 2) Through an analysis of the identifiability of CLIP, we theoretically prove the existence of a linear mapping from CLIP embeddings to the invariant factors. This mapping can be estimated using interventional data. Additionally, the OOD risk analysis provides a condition to guarantee a lower OOD risk for the invariant predictor. 3) We propose the CLIP-ICM framework, which includes collecting interventional data using either image or text data, estimating the linear projection, and performing invariant prediction. Extensive experiments across various OOD scenarios demonstrate that CLIP-ICM significantly enhances the performance of CLIP. Additionally, a comprehensive set of ablation studies validates the effectiveness of our approach.

## 2. Related Work

**Vision-Language Pre-training**. Incorporating language with the visual modality has been a long-standing issue, as extensively researched in previous works (Frome et al., 2013; Socher et al., 2013; Norouzi et al., 2014; Elhoseiny et al., 2013). Recent years have seen significant success in multi-modal pre-training models, spearheaded by CLIP (Radford et al., 2021) and ALIGN (Jia et al., 2021), with the aim of enabling cross-modal learning by mapping language and visual modalities into the same embedding space. Notably, researchers have turned their focus towards fine-tuning CLIP in downstream tasks (Gao et al., 2023; Zhou et al., 2022a;b; Shu et al., 2023). In our work, we delve into the reasons behind the poor OOD generalization ability of CLIP from a causal perspective. We propose CLIP-ICM as a solution to address this issue.

**Causal Representation Learning**. The latent factor hypothesis posits that observational data is generated by underlying latent factors, with causal representation learning aiming to identify these factors—a challenge akin to Independent Component Analysis (ICA) (Bishop, 1998; Bengio et al., 2013). While linear ICA is solvable (Hyvärinen & Oja, 2000), nonlinear ICA requires inductive biases such as auxiliary labels, sparsity constraints, or restricted function classes (Hyvärinen & Pajunen, 1999; Locatello et al., 2019; Schölkopf et al., 2021). Recent work has shifted from purely observational settings (Roeder et al., 2021) to leveraging interventional data (Ahuja et al., 2023; Squires et al., 2023). Aligning with this trend, we propose a method using intervention data to identify invariant features in CLIP, offering a simple yet effective solution.

**Out-of-distribution Generaliztion**. It is evident in many cases that the performance of deep learning methods is weakened when applied to different distributions (Beery et al., 2018; Taori et al., 2020). Studies have been conducted to address OOD generalization issues (Arjovsky et al., 2020; Ahuja et al., 2020; Gulrajani & Lopez-Paz; Miller et al., 2021; Abbe et al., 2023; Chen et al.). One important branch of work aims to find invariant causal mechanisms across domains, inspired by the invariance principle from causality (Arjovsky et al., 2020; Peters et al., 2016; Suter et al., 2019; Ahuja et al., 2020; 2021; Chen et al.). Additionally, some studies incorporate SCM when analyzing the OOD problem (Robey et al., 2021; Li et al., 2024). Vision-language pre-trained models, such as CLIP (Radford et al., 2021), exhibit impressive performance across various domains. However, recent works emphasize that adapting CLIP with task-specific data often comes at the cost of OOD generalization ability (Shu et al., 2023; Gao et al., 2023; Pham et al., 2023). We also provide a detailed analysis with some prior works in Section L.

# 3. Problem Setting

## 3.1. Contrastive Language-Image Pre-training

We begin by revisiting the framework for CLIP (Radford et al., 2021) [1]. Let $\boldsymbol{x} \in \mathcal{X}$ represent an arbitrary image, and $\boldsymbol{t} \in \mathcal{T}$ denote a text sequence describing this image, forming an image-text pair $(\boldsymbol{x}, \boldsymbol{t})$. Here, $\mathcal{X}$ and $\mathcal{T}$ refer to the image and text spaces, respectively. CLIP processes these inputs through two pre-trained encoders: an image encoder $f_I : \mathcal{X} \to \hat{\mathcal{Z}}$ and a text encoder $f_T : \mathcal{T} \to \hat{\mathcal{Z}}$. Both encoders map their respective inputs into a shared embedding space $\hat{\mathcal{Z}} \subseteq \mathbb{R}^D$, where the embeddings of the image-text pair $(\boldsymbol{x}, \boldsymbol{t})$ are aligned. Here, $D$ denotes the dimensionality of the embedding space.

A specific characteristic of CLIP is the zero-shot prediction. The goal of zero-shot prediction is to use a predictor $h : \mathcal{X} \to \mathcal{Y}$ to map an input image $\boldsymbol{x} \in \mathcal{X}$ to a predicted category label $\hat{y} \in \mathcal{Y}$, where $\mathcal{Y}$ denotes the set of possible class labels. For each class $c \in \mathcal{Y}$, a corresponding text description $\boldsymbol{t}_c$ is defined. For instance, for a class $c$, the description may be something like "a photo of a [CLASS]", where [CLASS] is replaced by the name of class $c$. CLIP encodes both the image $\boldsymbol{x}$ and the text description $\boldsymbol{t}_c$ for each class into the shared embedding space $\hat{\mathcal{Z}} \subseteq \mathbb{R}^D$. Then, it computes the similarity between the image embedding $f_I(\boldsymbol{x})$ and the embedding for the text description of each class $f_T(\boldsymbol{t}_c)$. The predictor $h$ is defined as:

$$h(\boldsymbol{x}) = \arg\max_{c \in \mathcal{Y}} P(c|\boldsymbol{x}),$$
$$\text{s.t.} \quad P(c|\boldsymbol{x}) = \frac{\exp\left(S(f_I(\boldsymbol{x}), f_T(\boldsymbol{t}_c))\right)}{\sum_{c' \in \mathcal{Y}} \exp\left(S(f_I(\boldsymbol{x}), f_T(\boldsymbol{t}_{c'}))\right)}. \quad (1)$$

Here, $P(c|\boldsymbol{x})$ is the probability of class $c$ given $\boldsymbol{x}$, $S(f_I(\boldsymbol{x}), f_T(\boldsymbol{t}_c))$ denotes the cosine similarity between $f_I(\boldsymbol{x})$ and $f_T(\boldsymbol{t}_c)$. The predicted label $\hat{y}$ is the one associated with the text description that has the highest similarity to the image embedding.

## 3.2. Out-Of-Distribution Generalization

Next, we present a unified framework for adapting CLIP to OOD generalization scenarios. In this framework, each distinct scenario (e.g., varying lighting, styles, or camera angles) is treated as an environment $e \in E_{all}$, where $E_{all}$ represents the set of all possible environments, and each environment corresponds to a unique data distribution $P^e(\boldsymbol{x}, y)$. The goal of OOD generalization is to learn a predictor $h$ from training environments $E_{tr} \subset E_{all}$ to perform well in unseen environments. Achieving this involves addressing two challenges that often occur together in practice: domain shift and open-class scenarios. Here, domain shift refers to when the distribution of $\boldsymbol{x}$ in the test environment dif-

fers from the distribution of $\boldsymbol{x}$ in the training environment[2], while open-class scenarios involve previously unseen classes appearing at test time. Although the zero-shot capability of CLIP excels in handling open-class scenarios, maintaining this ability under domain shift presents a challenge.

Several established methods exist for fine-tuning CLIP to acquire such a predictor $h$ for tasks in OOD scenarios: (1) Linear probe: A linear classifier is trained on the fixed output of $f_I$. (2) Learnable prompt (Zhou et al., 2022b;a; Zhang et al., 2024): The fixed text descriptions for each class are replaced with learnable parameters. (3) Adapter (Gao et al., 2023): Both $f_I$ and $f_T$ are frozen, and a multi-layer perceptron (MLP) is trained on their outputs. (4) Fine-tuning image encoder (Shu et al., 2023): All parameters of $f_I$ are optimized, while $f_T$ are fixed. Despite these differences, all methods share a common procedure when fine-tuned in OOD scenarios. A pre-trained CLIP model is taken as the backbone, and some task-specific parameters $\theta$, such as classifier weights or prompt vectors, are introduced. Data from the training environments $E_{tr} \subset E_{all}$ is then collected. During training, model parameters are optimized to reduce classification errors across the aggregated training distribution. After this phase, the model is evaluated on test environments $E_{all} \setminus E_{tr}$ using the updated parameters.

# 4. Motivation Experiment

In this section, we evaluate the performance of CLIP under OOD generalization using the Terra Incognita dataset (Beery et al., 2018), which contains identical object categories of wildlife species across diverse, in-the-wild environmental conditions. We first examine only the domain shift scenario, where we adopt a leave-one-out protocol and train a linear probe with the frozen CLIP image encoder. Specifically, for a given target domain, a linear classifier is trained on frozen CLIP image embeddings from all other domains and tested on the held-out domain to assess how well the model handles shifts in distribution. Next, we examine a case combining domain shift and open classes to assess whether the strong zero-shot capabilities of CLIP remain after fine-tuning in OOD settings. We divide categories into base and new classes and apply leave-one-out protocol. The image encoder is fine-tuned on base-class data from training domains, while the text encoder remains frozen. The model is then evaluated on both base and new classes in the target domain to assess whether the zero-shot ability of CLIP remains after fine-tuning.

From the results in Table 1, we can draw two important conclusions. First, in the domain shift scenario, the performance of fine-tuning is unsatisfactory. For instance, in the

---

[1]The definition of all notations is provided in Appendix A.

[2]We primarily refer to the case in standard domain generalization tasks.

*Table 1.* The performance of CLIP on the Terra Incognita dataset in accuracy (%). L100, L38, L43, and L46 represent different environments. In the Linear-Probe case, the values in ($\cdot$) represent the accuracy achieved by directly fine-tuning with the linear probe on the target domain.

| | **DOMAIN SHIFT** | | | | |
|---|---|---|---|---|---|
| METHOD | L100 | L38 | L43 | L46 | AVG |
| LINEAR-PROBE | 73.6 (90.9) | 58.3 (86.3) | 61.0 (76.0) | 47.8 (78.9) | 60.2 (83.0) |
| | **OPEN CLASS UNDER DOMAIN SHIFT** | | | | |
| SPLIT | METHOD | L100 | L38 | L43 | L46 | AVG |
| BASE | ZERO-SHOT | 56.4 | 23.2 | 30.4 | 25.8 | 33.9 |
| | FINE-TUNE | 75.6 | 58.5 | 65.2 | 55.1 | 63.6 |
| NEW | ZERO-SHOT | 47.6 | 17.6 | 35.2 | 37.4 | 34.5 |
| | FINE-TUNE | 36.7 | 11.2 | 25.1 | 25.5 | 24.6 |
| TOTAL | ZERO-SHOT | 52.0 | 20.4 | 32.8 | 31.6 | 34.2 |
| | FINE-TUNE | 56.2 | 34.9 | 45.2 | 40.3 | 44.2 |

L46 domain, the accuracy achieved using the leave-one-out protocol (47.8%) is significantly lower by 31.1% compared to directly fine-tuning on this domain (78.9%). This indicates that training on other domains does not ensure good performance on the L46 domain. Second, in the open-class scenario, the zero-shot performance remains nearly consistent across base and new classes. While the fine-tuned model outperforms zero-shot predictions on base classes, it performs worse on new classes. These results suggest that naively fine-tuning the CLIP model in OOD scenarios can harm its strong zero-shot capabilities. This raises a critical question: What causes the discrepancy between training and testing domains in the OOD generalization of CLIP, and how can we preserve the strong zero-shot capacity of CLIP for unseen classes while adapting to OOD scenarios?

## 5. Theoretical Analysis

### 5.1. Causal Analysis

To investigate the reasons behind the unsatisfactory OOD generalization of CLIP further, we propose to analyze it from a causal perspective. During analysis, we adopt the latent factor hypothesis, which assumes that a set of latent factors generates the observed data. Based on this, we model both the data-generating and prediction processes using a Structural Causal Model (SCM) in Figure 1 [3].

Let $X \sim P(\boldsymbol{x})$ and $Y \sim P(y)$ be the random variables for images and class labels, defined on $\mathcal{X}$ and $\mathcal{Y}$, respectively. A selection variable $E \sim P(e)$, defined on $E_{all}$, represents the environment (Pearl & Bareinboim, 2014). The latent

---

[3]Please refer to Definition C.1 for the definition of the SCM and the causal mechanism.

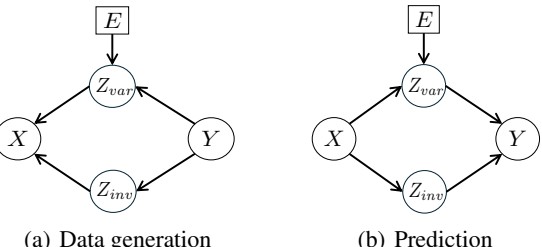

(a) Data generation    (b) Prediction

*Figure 1.* The SCMs for: (a) the data-generating process (b) the prediction process. The square denotes the selection variable, see Appendix C for details.

variable $Z \sim P(\boldsymbol{z})$, defined on $\mathcal{Z} = \mathcal{Z}_{inv} \times \mathcal{Z}_{var}$, consists of two components: the invariant factor $Z_{inv} \sim P(\boldsymbol{z}_{inv})$, and the variant factor $Z_{var} \sim P(\boldsymbol{z}_{var})$.

The SCM in Figure 1(a) illustrates the data-generating process. In this SCM, the edges $Y \to Z_{inv}$ and $Y \to Z_{var}$ indicate that the class label $Y$ influences both $Z_{inv}$ and $Z_{var}$. The edge $E \to Z_{var}$ shows that $Z_{var}$ is affected by the environment $E$. Finally, the edges $Z_{inv} \to X$ and $Z_{var} \to X$ represent that the image $X$ is generated from $Z_{inv}$ and $Z_{var}$. For example, consider the class label "bird" ($Y$). The wing shape and beak shape ($Z_{inv}$) do not change across different environments. However, feather color ($Z_{var}$) appears differently in day or night ($E$). Resulting in varying images of birds ($X$) across environments.

The SCM in Figure 1(b) depicts the prediction process, which can be considered as the inverse of the data-generating process (Bengio et al., 2013; Schölkopf et al., 2021; Zimmermann et al., 2021). In this SCM, the edges $X \to Z_{var}$ and $X \to Z_{inv}$ denote the representation learn-

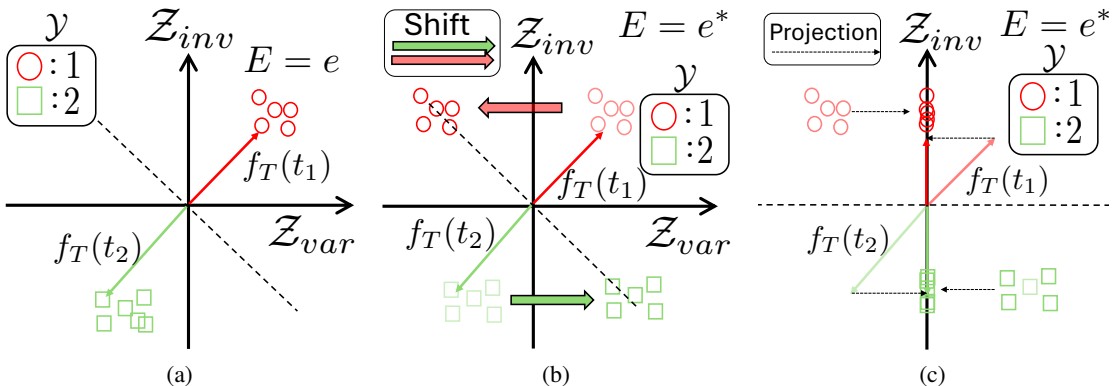

Figure 2. An example of a classifier defined on $\mathcal{Z} = \mathcal{Z}_{inv} \times \mathcal{Z}_{var}$ when applied to different environments, where the dashed lines represent the decision boundary and solid arrows depict the normal vectors of these boundaries. $\mathcal{Y}$ is the set of labels. (a) illustrates an example of a classifier that is trained and tested in the same environment. (b) demonstrates applying the classifier from (a) to another environment $E = e^*$. The change in $E$ only affects the distribution of $Z_{var}$. (c) demonstrate the case when samples and classifiers are projected to the $\mathcal{Z}_{inv}$ space.

ing process. The edges $Z_{var} \to Y$ and $Z_{inv} \to Y$ represent the prediction process. In OOD prediction scenarios of CLIP, the process $X \to Z_{inv}$ and $X \to Z_{var}$ correspond to the image being passed through the image encoder to obtain its feature representation, i.e., $f_I(x) = z$, where $z = [z_{inv}, z_{var}]^T$ [4]. The process $Z_{inv} \to Y$ and $Z_{var} \to Y$ involve comparing the image embedding $z$ with the embeddings of text descriptions $f_T(t_c)$ for each class $c \in \mathcal{Y}$, and the predicted label is derived based on the similarity, as shown in Equation (1).

In an SCM, each edge represents a specific causal mechanism. Our focus is the causal mechanism in the OOD prediction scenario. Specifically, we investigate whether a predictor learned from training environment $E_{tr} \subset E_{all}$ can still perform well in an unknown test environment $e^* \in E_{all}$. The following proposition outlines how the causal mechanisms change under environmental shifts:

**Proposition 5.1.** *Let $P^*(\cdot) := P(\cdot|e^*)$ denote the distribution in test environment $E = e^*$. The causal mechanisms are formalized through interventional distributions. We focus on the predictions $P(y|do(\cdot))$, where $do(\cdot)$ represents an intervention. The causal mechanism satisfies $P^*(y|do(x)) \neq P(y|do(x))$. When using only $Z_{inv}$, the causal mechanism remains unchanged: $P^*(y|do(z_{inv})) = P(y|do(z_{inv}))$.*

The detailed proof is provided in Appendix D. Proposition 5.1 shows that the causal mechanism $P(y|do(x))$, relying on both invariant and variant factors, varies across environments, making a predictor trained in one domain unusable in another. In contrast, the causal mechanism $P(y|do(z_{inv}))$, based only on $Z_{inv}$, remains stable across

environments, ensuring a predictor trained in one domain can generalize to others.

To illustrate this process, Figure 2 provides an example. In the figure, we plot the image samples in the latent space $\mathcal{Z} = \mathcal{Z}_{inv} \times \mathcal{Z}_{var}$. Samples from different classes are plotted in different colors. The arrows in varying colors represent the embedding of text descriptions for each class. Figure 2(a) shows the sample distribution in train environment $e$, while Figure 2(b) depicts the distribution in test environment $e^*$. In Figure 2(a), the samples of a given color are similar to the corresponding text description features, ensuring correct predictions in the training environment.[5] However, in Figure 2(b), the distribution of samples shifts along the $Z_{var}$ dimension due to environmental changes. This shift causes samples to move away from their corresponding text embeddings, leading to misclassification. In Figure 2(c), both the samples and text embeddings are projected into the $\mathcal{Z}_{inv}$ subspace. Here, the environmental shift no longer affects the prediction. This ensures a consistent performance in both train and test environments.

## 5.2. Identifiability Analysis

In the previous section, we propose that if the prediction is performed in the space $\mathcal{Z}_{inv}$, the results in test environment remain consistent with training environments. However, in practice, it is unclear which part of the image representation obtained by CLIP corresponds to $\mathcal{Z}_{inv}$. The task of finding invariant factors mirrors the well-established problem of latent factor identification (Schölkopf et al., 2021; Hyvärinen & Oja, 2000). The observational data are assumed to be generated by the latent factor $Z$ through an unknown,

---

[4]For simplicity, this process assumes an ideal scenario, which we discuss further in Section 5.2.

[5]This remains true regardless of the fine-tuning strategy, even in open-class scenarios. For further discussion, refer to Appendix B.

non-linear process $g$. Identifying the latent factors meaning finding a function $f : \mathcal{X} \to \mathcal{Z}$ such that $f$ is the inverse of data generation process $g : \mathcal{Z} \to \mathcal{X}$, i.e., $f = g^{-1}$. In this section, we theoretically prove that it is possible to find a linear mapping from the CLIP embedding space to $\mathcal{Z}_{inv}$.

We aim to explore the ability of CLIP to identify latent variables. To begin with, we assume the data generation process $g : \mathcal{Z} \to \mathcal{X}$ is injective. This assumption implies that $g$ is invertible, and different latent factors generate different images. Based on this generation process, we define the ideal image encoder $f_{I*} : \mathcal{X} \to \mathcal{Z}$ as the inverse of $g$, i.e. $f_{I*}(\boldsymbol{x}) = g^{-1}(\boldsymbol{x})$. Furthermore, we define $f_{T*}$ as the corresponding text encoder for $f_{I*}$. The outputs of pairs $(\boldsymbol{x}, \boldsymbol{t})$ through $f_{T*}$ and $f_{I*}$ are aligned in the shared embedding space. For the actual encoders $f_I$ and $f_T$ in CLIP, we prove in Proposition 5.3 that if Condition 5.2 is satisfied, $f_I$ identifies $\mathcal{Z}$ up to an invertible linear transformation.

**Condition 5.2.** *For an image encoder $f_I$ and a text encoder $f_T$ with output dimensions $D$, for any $\boldsymbol{x}$ sampled from $P(\boldsymbol{x})$, there exist $D + 1$ distinct text description pairs $(\boldsymbol{t}_a, \boldsymbol{t}_b)$ satisfying:*

$$\frac{\exp(f_I(\boldsymbol{x})^T f_T(\boldsymbol{t}_a))}{\exp(f_I(\boldsymbol{x})^T f_T(\boldsymbol{t}_b))} = \frac{\exp(f_{I*}(\boldsymbol{x})^T f_{T*}(\boldsymbol{t}_a))}{\exp(f_{I*}(\boldsymbol{x})^T f_{T*}(\boldsymbol{t}_b))}. \quad (2)$$

**Proposition 5.3.** *For an image encoder $f_I : \mathcal{X} \to \hat{\mathcal{Z}}$ and its corresponding text encoder $f_T : \mathcal{T} \to \hat{\mathcal{Z}}$, where $\hat{\mathcal{Z}} \subseteq \mathbb{R}^D$ is the embedding space. If Condition 5.2 is satisfied, then $f_I(\boldsymbol{x})$ identifies $\mathcal{Z}$ up to an invertible linear transformation $A$, i.e. $f_I(\boldsymbol{x}) = Ag^{-1}(\boldsymbol{x}) = A\boldsymbol{z}$.*

The proof is provided in Appendix E[6]. Condition 5.2 ensures both **consistency** and **diversity**. Consistency is achieved as the similarity between image embeddings and text embeddings produced by $f_I$ and $f_T$ aligns with those produced by the ideal encoders $f_{I*}$ and $f_{T*}$. Diversity is ensured through the existence of sufficient pairs $(\boldsymbol{t}_a, \boldsymbol{t}_b)$, which guarantees the invertibility of the matrix $A$. Consequently, satisfying Proposition 5.3 ensures that predictions based on similarity remain consistent with the ideal setting, allowing CLIP to achieve accurate predictions. The remarkable performance of CLIP, supported by its large-scale training data, suggests that it can satisfy this condition in practice.

Proposition 5.3 suggests that the output representation of CLIP is a linear combination of the true latent space, encompassing both invariant and variant factors. As a result, changes in the environment lead to shifts in all dimensions of the output of CLIP. As we discussed in Section 5.1, this can lead to incorrect prediction when the text descrip-

tion remains unchanged. Fortunately, **Proposition 5.3 indicates there exists a linear mapping from $\hat{\mathcal{Z}}$ to $\mathcal{Z}_{inv}$.** Consider an observed representation $\hat{\boldsymbol{z}} = f_I(\boldsymbol{x})$, Proposition 5.3 suggests $\hat{\boldsymbol{z}}$ is a linear transformation of the true latent vector, i.e., $\hat{\boldsymbol{z}} = A\boldsymbol{z}$, where $A$ is an invertible matrix, $\boldsymbol{z} = [\boldsymbol{z}_{inv}, \boldsymbol{z}_{var}]^T$ is the true latent vector. The true latent vector can be recovered by applying the inverse of $A$, so we have $\boldsymbol{z} = A^{-1}\hat{\boldsymbol{z}}$. We can decompose $A^{-1}$ into two blocks $A^{-1} = \begin{bmatrix} A_{inv} & A_{var} \end{bmatrix}^T$. And $\boldsymbol{z}_{inv}$ can be obtained with $\boldsymbol{z}_{inv} = A_{inv}\hat{\boldsymbol{z}}$. Since $A$ is a fixed, invertible matrix, the above derivative shows that there always exists a linear mapping $A_{inv}$ that maps $\hat{\boldsymbol{z}}$ to the true invariant factor $\boldsymbol{z}_{inv}$.

Since both $A$ and the true latent factors $\boldsymbol{z}$ are unobservable. It is unlikely to directly obtain the matrix $A_{inv}$ through only the observed representation $\hat{\boldsymbol{z}}$. However, by using interventional data, where certain latent factors are fixed, as outlined in Condition 5.4, we can prove that it becomes possible to estimate the $A_{inv}$.

**Condition 5.4.** *Let $P^{do(z_{inv})}(\boldsymbol{x})$ denote an interventional distribution of $X$. The invariant factors of the images from this distribution are required to take a fixed value, i.e. $z_{inv} = z^\dagger$. Here, $do(\cdot)$ denotes an intervention flag. This condition require sampling from $P^{do(z_{inv})}(\boldsymbol{x})$ is available.*

**Proposition 5.5.** *For the encoder $f_I$ satisfies Proposition 5.3, if Condition 5.4 is satisfied and sample from $P^{do(z_{inv})}(\boldsymbol{x})$ is possible. Then for any pair $(\boldsymbol{x}_1^{do(z_{inv})}, \boldsymbol{x}_2^{do(z_{inv})})$ sampled from $P^{do(z_{inv})}(\boldsymbol{x})$. The mapping $A_{inv}$ satisfies:*

$$A_{inv}(f_I(\boldsymbol{x}_1^{do(z_{inv})}) - f_I(\boldsymbol{x}_2^{do(z_{inv})})) = 0 \quad (3)$$

The proof is detailed in Appendix F. Proposition 5.5 provides the theoretical foundation for extracting $\boldsymbol{z}_{inv}$ and performing prediction in the $\mathcal{Z}_{inv}$ space. Once Condition 5.4 is satisfied, interventional data can be obtained. By repeatedly sampling from this interventional distribution, it becomes possible to estimate a matrix that satisfies Equation (3). This matrix then maps the observable data into the $\mathcal{Z}_{inv}$ space, allowing an invariant prediction across environments.

### 5.3. OOD Generalization Analysis

In the previous sections, we prove the feasibility of projecting representations of CLIP into an invariant subspace and performing predictions within that subspace. In this section, we provide theoretical guarantees for the OOD generalization performance of this approach.

One way to assess OOD generalization is to measure performance in the worst-case environment. Formally, this is captured by the OOD risk:

$$R^{\text{OOD}}(h) = \max_{e \in E_{all}} R^e(h), \quad (4)$$

---

[6]This conclusion aligns with results established in prior works (Roeder et al., 2021; Ahuja et al., 2022; Hyvarinen & Morioka, 2016; Zimmermann et al., 2021), and Proposition 5.3 demonstrates this result within our specific scenario.

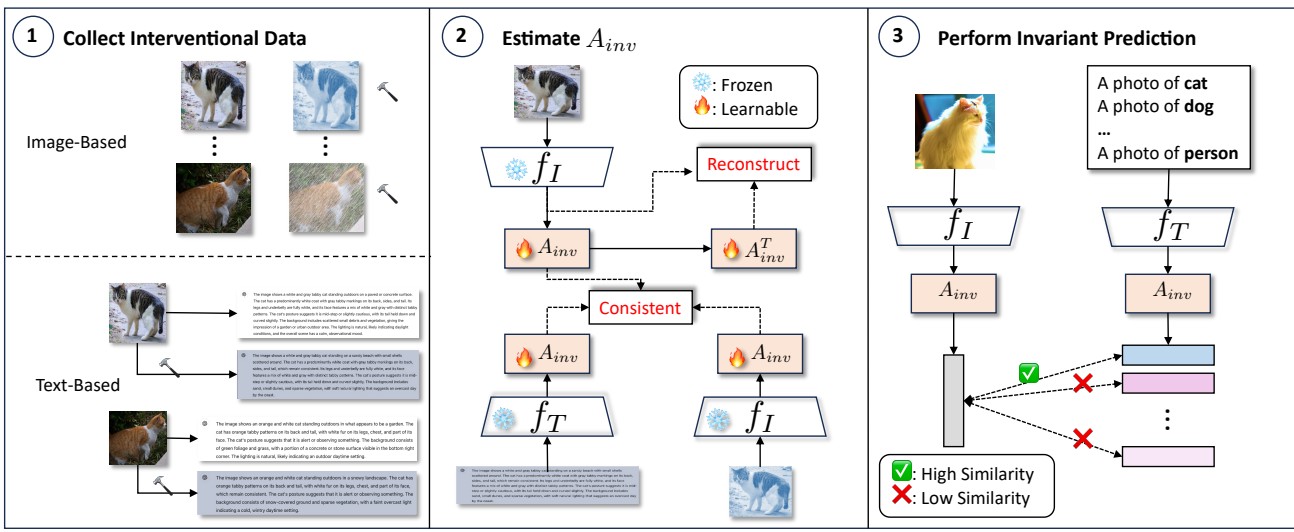

*Figure 3.* The pipeline of the proposed CLIP-ICM framework. CLIP-ICM consists of three stages: (1) collect interventional data. (2) estimate $A_{inv}$. (3) perform invariant prediction.

where $R^e(h)$ denotes the risk of $h$ in environment $e$, and can be calculated with:

$$R^e(h) = \mathbb{E}_{(P^e(\boldsymbol{x}, y)}\left[\mathbb{I}(h(\boldsymbol{x}) \neq y)\right], \quad (5)$$

where $\mathbb{I}(h(\boldsymbol{x}) \neq y)$ is the indicator function. It equals 1 if the prediction of $h$ is incorrect and 0 otherwise.

We consider two types of predictors: the conventional predictor $h$ and the invariant predictor $h_{inv}$. The definition of the conventional predictor is provided in Equation (1), while the invariant predictor is defined as follows:

$$h_{inv}(\boldsymbol{x}) = \arg\max_{c \in \mathcal{Y}} P(c|\boldsymbol{x}, A_{inv}), \quad (6)$$

where $P(c|\boldsymbol{x}, A_{inv})$ is the probability of class $c$ computed in the invariant subspace. It can be obtained by projecting with $A_{inv}$ and applying the following softmax function:

$$P_{invk}(c|\boldsymbol{x}) = \frac{\exp\left(S(A_{inv}f_I(\boldsymbol{x}), A_{inv}f_T(\boldsymbol{t}_c))\right)}{\sum_{c' \in \mathcal{Y}} \exp\left(S(A_{inv}f_I(\boldsymbol{x}), A_{inv}f_T(\boldsymbol{t}_{c'}))\right)}. \quad (7)$$

By investigating the OOD risk of these predictors, we present the following theorem:

**Theorem 5.6.** *Let $\mathcal{H}$ be a hypothesis class over $\mathcal{X} \times \mathcal{Y}$, and let $E_{all}$ denote a set of environments with distributions $\{P^e\}_{e \in E_{all}}$ over $\mathcal{X} \times \mathcal{Y}$. Let $h_{inv} \in \mathcal{H}$ be a predictor relying solely on $Z_{inv}$, and $h \in \mathcal{H}$ a predictor utilizing both $Z_{inv}$ and $Z_{var}$. If the mutual information between $Z_{inv}$ and $Z$ satisfies $I(Z_{inv}; Z) > c$ for some constant $c > 0$. Then, the worst-case OOD risk strictly satisfies:*

$$R^{\text{OOD}}(h_{inv}) < R^{\text{OOD}}(h), \quad (8)$$

The proof is detailed in Appendix G. Theorem 5.6 state that, to ensure a smaller OOD risk, one needs to ensure a high $I(Z_{inv}; Z)$. This means $Z_{inv}$ should contain enough information related to $Z$, thereby ensuring that the invariant predictor also performs well in the training environment.

## 6. Method

Building on the theoretical results in Section 5, we propose *Invariant Causal Mechanism of CLIP* (CLIP-ICM). CLIP-ICM comprises three stages: (1) Collect interventional data. (2) Estimating $A_{inv}$ using the interventional data. (3) Perform prediction in the invariant subspace. The key challenges lie in two aspects: acquiring interventional data and estimating $A_{inv}$ satisfying the condition in both Theorem 5.6 and Proposition 5.5.

**Collect Interventional data**. According to Condition 5.4, interventional data are those generated by fixing the invariant factors while allowing the variant factors to vary. There are numerous ways to obtain such data (Nikolenko, 2021), including the use of rendering engines (Zhang et al., 2025), generative models (Zhang et al., 2023a; Azizi et al.), or large language models (Li et al., 2023; OpenAI, 2023). Leveraging the property of CLIP to align image and text features within the same embedding space, we propose two approaches for collecting interventional data: one based on images and the other based on text.

We first collect image data from a subset of training environments $E_{tr} \subset E_{all}$. The interventional data is generated by applying data augmentation techniques. Data augmentation preserves $Z_{inv}$ while altering $Z_{var}$. This works because

invariant factors, like class-relevant features, remain unchanged, while variant factors, such as color, orientation, or background, are modified. We use a combination of methods, including color jittering, grayscale, and Gaussian blur, to create diverse transformations. Details on these techniques are provided in Appendix H, and an ablation study of specific augmentations is discussed in Appendix M.1. For each image $x$, we apply a random augmentation $\alpha(\cdot)$ to generate an augmented sample $\alpha(x)$. Embeddings of the pair $f_I(\alpha(x))$, $f_I(x)$ are then used as interventional data.

Due to the ability of CLIP to align the image and text embedded in the same latent space, interventional data can also be generated by modifying the text descriptions associated with a given image. We use the same training images from $E_{tr}$. Pre-intervention text descriptions are generated for each image using an image captioning model. Post-intervention text descriptions are created by modifying these captions using a large language model. Details of the prompts used for generating and modifying text descriptions are provided in Appendix H.1. Finally, the embeddings of the pre- and post-intervention text descriptions, $f_T(t)$ and $f_T(\beta(t))$, are used as interventional data for further processing.

**Estimate** $A_{inv}$. Once the interventional data is obtained, the next step is to learn the $A_{inv}$ matrix. The $A_{inv}$ matrix must satisfy two key conditions: (1) According to Proposition 5.5, $A_{inv}$ should satisfy Equation (3), ensuring that the embeddings obtained from interventional data pairs are equal after applying $A_{inv}$. (2) Based on Theorem 5.6, the projected embedding should retain as much information about the original representation $f_I(x)$ as possible. To meet these conditions, we formulate the learning objective as the following optimization problem:

$$\min_{A_{inv}} \|\hat{Z} - A_{inv}A_{inv}^T\hat{Z}\|_F^2 + \lambda\|A_{inv}^T(\hat{Z} - \hat{Z}^{do(z_{inv})})\|_F^2,$$
$$\text{s.t.} \quad A_{inv}^T A_{inv} = I_{D_{inv}}. \tag{9}$$

Here, $\hat{Z}$ represents the matrix of observable embeddings obtained from either the image encoder $f_I$ or the text encoder $f_T$. $\hat{Z}^{do(z_{inv})}$ corresponds to the embeddings obtained from interventional data, where the invariant factors $Z_{inv}$ are fixed while the variant factors $Z_{var}$ vary. $I_{D_{inv}}$ denotes the identity matrix of dimension $D_{inv}$, and $\lambda$ is a regularization hyperparameter. The first term of Equation (9) along with the constraint $A_{inv}^T A_{inv} = I_{D_{inv}}$ ensures that $\hat{Z}$ is effectively reconstructed within the invariant subspace defined by $A_{inv}$. The second term enforces consistency between the embeddings of interventional data in the invariant subspace.

**Perform invariant prediction.** After obtaining $A_{inv}$ through Equation (9), we can perform invariant prediction. For a test image $x$ and a fixed set of text prompts $\{t_c\}_{c \in \mathcal{Y}}$, the embeddings are first computed using $f_I$ and $f_T$, respectively. These embeddings are then projected into the invariant subspace using $A_{inv}$. Predictions are made using the invariant predictor defined in Equation (6). [7]

# 7. Experiment

To evaluate the CLIP-ICM framework in OOD scenarios, we conduct experiments on the DomainBed benchmark (Gulrajani & Lopez-Paz). We use five datasets from DomainBed: PACS (Li et al., 2017), VLCS (Fang et al., 2013), Office-Home (Venkateswara et al., 2017), Terra Incognita (Beery et al., 2018), and DomainNet (Peng et al., 2019).[8]

We evaluate CLIP-ICM for OOD generalization in two settings: (1) domain shift and (2) domain shift with open-class scenarios. For domain shift, we use a leave-one-out protocol, training on all domains except the target and testing on the target domain (Table 2). For the combined setting, we split data into base and new classes, train on base classes in training domains, and evaluate both base and new classes in the target domain (Table 3). Each value in Table 2 and Table 3 represents the mean and standard deviation over 5 runs with different random seeds.

We compare three interventional data collection methods: models marked with * use image-based data, † use text-based data, and unmarked use both. We also compare two prediction strategies: CLIP-ICM, which maps embeddings through $A_{inv}$, and CLIP-ICM Linear Probe, which trains a classifier on image embeddings projected by $A_{inv}$.

**Results.** From Table 2, CLIP-ICM consistently outperforms previous fine-tuning methods for CLIP. On the Terra Incognita dataset, CLIP-ICM Linear Probe surpasses the best-performing method, CLIPOOD, by 6%. From Table 3, while CLIP-ICM does not achieve the best performance on base classes, it significantly outperforms other methods on new classes. The average performance on new classes exceeds the best-performing method, CoOp, by 5.5%. This shows that the CLIP-ICM learning method preserves the zero-shot capability of CLIP even after fine-tuning. Comparing the three approaches for interventional data, models using only image-based data outperform those using only text-based data. Combining both further improves performance, highlighting their complementary benefits.

# 8. Conclusion

In our work, we observe the limitations of CLIP in OOD scenarios. Causal analysis reveals that predictors relying

---

[7]Alternatively, predictions can be made by projecting only the image embeddings through $A_{inv}$ and training a classifier on the projected space. This approach is more suitable when the classes in the training environments are identical to those in the test environments (i.e., no open-class scenarios).

[8]See Appendices I, J and O for additional results and Appendix M for ablation studies.

*Table 2.* Accuracy on the DomainBed benchmark with domain shift. Methods with * indicate training with only image-based interventional data. Methods with † indicate training with only text-based interventional data.

| METHOD | BACKBONE | PACS | VLCS | OFFICEHOME | TERRAINC | DOMAINNET | AVG. |
|---|---|---|---|---|---|---|---|
| ZERO-SHOT | CLIP | 96.1 | 82.4 | 71.5 | 34.2 | 56.8 | 68.2 |
| LINEAR-PROBE | CLIP | $96.4_{\pm0.1}$ | $78.7_{\pm0.2}$ | $81.9_{\pm0.4}$ | $60.2_{\pm0.2}$ | $55.0_{\pm0.4}$ | $74.4_{\pm0.4}$ |
| MIRO (CHA ET AL., 2022) | CLIP | $95.6_{\pm0.2}$ | $82.2_{\pm0.1}$ | $82.5_{\pm0.3}$ | $54.3_{\pm0.2}$ | $54.0_{\pm0.5}$ | $73.7_{\pm0.5}$ |
| COOP (ZHOU ET AL., 2022B) | CLIP | $97.0_{\pm0.2}$ | $83.0_{\pm0.1}$ | $81.1_{\pm0.4}$ | $54.6_{\pm0.2}$ | $59.5_{\pm0.2}$ | $75.0_{\pm0.2}$ |
| COCOOP (ZHOU ET AL., 2022A) | CLIP | $96.7_{\pm0.4}$ | $83.6_{\pm0.1}$ | $80.7_{\pm0.1}$ | $56.2_{\pm0.3}$ | $59.7_{\pm0.4}$ | $75.4_{\pm0.4}$ |
| CLIP-ADAPTER (GAO ET AL., 2023) | CLIP | $96.4_{\pm0.3}$ | $84.3_{\pm0.5}$ | $82.2_{\pm0.2}$ | $57.5_{\pm0.4}$ | $59.9_{\pm0.1}$ | $76.1_{\pm0.1}$ |
| DPL (ZHANG ET AL., 2023B) | CLIP | $97.3_{\pm0.5}$ | $84.3_{\pm0.1}$ | $84.2_{\pm0.2}$ | $52.6_{\pm0.4}$ | $56.7_{\pm0.4}$ | $75.0_{\pm0.4}$ |
| CLIPOOD | CLIP | $97.3_{\pm0.1}$ | $85.0_{\pm0.4}$ | $87.0_{\pm0.2}$ | $60.4_{\pm0.7}$ | $63.5_{\pm0.4}$ | $78.6_{\pm0.1}$ |
| CLIP-ICM* | CLIP | $97.3_{\pm0.5}$ | $84.1_{\pm0.4}$ | $82.6_{\pm0.3}$ | $49.9_{\pm0.3}$ | $60.5_{\pm0.3}$ | $74.9_{\pm0.4}$ |
| CLIP-ICM* LINEAR-PROBE | CLIP | $97.5_{\pm0.5}$ | $86.5_{\pm0.1}$ | $84.6_{\pm0.4}$ | $64.3_{\pm0.3}$ | $64.0_{\pm0.2}$ | $79.0_{\pm0.3}$ |
| CLIP-ICM† | CLIP | $96.8_{\pm0.4}$ | $83.4_{\pm0.3}$ | $82.1_{\pm0.4}$ | $45.2_{\pm0.3}$ | $57.4_{\pm0.2}$ | $73.0_{\pm0.3}$ |
| CLIP-ICM† LINEAR-PROBE | CLIP | $97.2_{\pm0.5}$ | $85.2_{\pm0.5}$ | $82.4_{\pm0.3}$ | $61.2_{\pm0.1}$ | $59.6_{\pm0.4}$ | $77.1_{\pm0.4}$ |
| CLIP-ICM | CLIP | $97.7_{\pm0.2}$ | $86.2_{\pm0.3}$ | $84.6_{\pm0.2}$ | $52.5_{\pm0.4}$ | $61.1_{\pm0.3}$ | $76.4_{\pm0.3}$ |
| CLIP-ICM LINEAR-PROBE | CLIP | $\mathbf{97.8}_{\pm0.3}$ | $\mathbf{86.6}_{\pm0.1}$ | $\mathbf{87.1}_{\pm0.4}$ | $\mathbf{66.5}_{\pm0.1}$ | $\mathbf{65.0}_{\pm0.1}$ | $\mathbf{80.6}_{\pm0.2}$ |

*Table 3.* Accuracy on OfficeHome and DomainNet with both domain shift and open classes. Methods with * indicate training with only image-based interventional data. Methods with † indicate training with only text-based interventional data.

| SPLIT | METHOD | OFFICEHOME | | | | DOMAINNET | | | | | |
|---|---|---|---|---|---|---|---|---|---|---|---|
| | | A | C | P | R | C | I | P | Q | R | S |
| BASE | CLIP | 86.8 | 75.5 | 89.5 | 92.6 | 72.8 | 51.7 | 66.0 | 13.5 | 83.4 | 66.9 |
| | COOP | $87.0_{\pm0.4}$ | $78.3_{\pm1.2}$ | $92.4_{\pm0.2}$ | $91.4_{\pm0.6}$ | $75.7_{\pm0.2}$ | $58.8_{\pm0.5}$ | $68.5_{\pm1.3}$ | $13.1_{\pm1.0}$ | $84.0_{\pm0.5}$ | $70.0_{\pm0.1}$ |
| | CLIPOOD | $\mathbf{90.1}_{\pm0.2}$ | $\mathbf{79.7}_{\pm0.2}$ | $\mathbf{93.1}_{\pm0.1}$ | $\mathbf{94.8}_{\pm0.1}$ | $\mathbf{79.0}_{\pm0.2}$ | $\mathbf{62.2}_{\pm0.1}$ | $\mathbf{73.0}_{\pm0.2}$ | $\mathbf{20.2}_{\pm0.2}$ | $\mathbf{86.2}_{\pm0.1}$ | $\mathbf{73.8}_{\pm0.1}$ |
| | CLIP-ICM* | $88.6_{\pm0.4}$ | $78.0_{\pm0.2}$ | $90.2_{\pm0.3}$ | $93.1_{\pm0.2}$ | $74.4_{\pm0.3}$ | $53.2_{\pm0.3}$ | $67.2_{\pm0.3}$ | $14.6_{\pm0.2}$ | $85.2_{\pm0.3}$ | $67.8_{\pm0.2}$ |
| | CLIP-ICM† | $87.1_{\pm0.1}$ | $77.2_{\pm0.2}$ | $89.3_{\pm0.5}$ | $92.0_{\pm0.2}$ | $73.1_{\pm0.2}$ | $52.0_{\pm0.1}$ | $66.1_{\pm0.1}$ | $14.1_{\pm0.1}$ | $83.8_{\pm0.4}$ | $66.8_{\pm0.4}$ |
| | CLIP-ICM | $89.2_{\pm0.4}$ | $78.6_{\pm0.4}$ | $90.6_{\pm0.1}$ | $93.7_{\pm0.2}$ | $75.0_{\pm0.3}$ | $53.9_{\pm0.3}$ | $67.6_{\pm0.4}$ | $14.9_{\pm0.3}$ | $85.9_{\pm0.3}$ | $68.4_{\pm0.4}$ |
| NEW | CLIP | 76.6 | 59.4 | 88.1 | 86.2 | 70.2 | 44.1 | 66.4 | 14.1 | 83.5 | 61.0 |
| | COOP | $76.5_{\pm1.1}$ | $56.6_{\pm2.4}$ | $88.0_{\pm1.9}$ | $86.8_{\pm0.7}$ | $71.5_{\pm0.2}$ | $47.2_{\pm0.3}$ | $67.3_{\pm0.7}$ | $14.8_{\pm0.7}$ | $83.7_{\pm0.7}$ | $63.1_{\pm0.3}$ |
| | CLIPOOD | $77.8_{\pm0.2}$ | $60.0_{\pm0.2}$ | $88.3_{\pm0.1}$ | $86.7_{\pm0.1}$ | $71.2_{\pm0.1}$ | $48.1_{\pm0.1}$ | $68.2_{\pm0.2}$ | $18.0_{\pm0.4}$ | $83.4_{\pm0.1}$ | $62.9_{\pm0.1}$ |
| | CLIP-ICM* | $81.7_{\pm0.4}$ | $66.5_{\pm0.5}$ | $90.2_{\pm0.4}$ | $90.6_{\pm0.4}$ | $76.7_{\pm0.2}$ | $50.9_{\pm0.2}$ | $69.1_{\pm0.5}$ | $17.2_{\pm0.5}$ | $83.6_{\pm0.5}$ | $67.7_{\pm0.5}$ |
| | CLIP-ICM† | $81.2_{\pm0.2}$ | $65.2_{\pm0.2}$ | $89.4_{\pm0.4}$ | $89.9_{\pm0.2}$ | $76.0_{\pm0.2}$ | $49.8_{\pm0.1}$ | $67.9_{\pm0.2}$ | $15.7_{\pm0.1}$ | $82.5_{\pm0.1}$ | $67.0_{\pm0.4}$ |
| | CLIP-ICM | $\mathbf{82.6}_{\pm0.1}$ | $\mathbf{67.5}_{\pm0.2}$ | $\mathbf{90.9}_{\pm0.3}$ | $\mathbf{91.5}_{\pm0.3}$ | $\mathbf{77.8}_{\pm0.1}$ | $\mathbf{51.6}_{\pm0.5}$ | $\mathbf{70.2}_{\pm0.2}$ | $18.0_{\pm0.4}$ | $\mathbf{84.5}_{\pm0.2}$ | $\mathbf{68.6}_{\pm0.5}$ |
| TOTAL | CLIP | 82.6 | 67.3 | 88.8 | 89.5 | 71.4 | 47.1 | 66.2 | 13.8 | 83.4 | 63.4 |
| | COOP | $82.7_{\pm0.5}$ | $67.2_{\pm0.7}$ | $90.2_{\pm1.0}$ | $89.2_{\pm0.6}$ | $73.4_{\pm0.3}$ | $51.8_{\pm0.3}$ | $67.9_{\pm1.0}$ | $13.7_{\pm0.8}$ | $83.9_{\pm0.5}$ | $66.0_{\pm0.2}$ |
| | CLIPOOD | $85.1_{\pm0.1}$ | $69.6_{\pm0.2}$ | $90.8_{\pm0.1}$ | $91.0_{\pm0.1}$ | $74.8_{\pm0.1}$ | $\mathbf{53.6}_{\pm0.1}$ | $\mathbf{70.6}_{\pm0.1}$ | $\mathbf{19.1}_{\pm0.3}$ | $84.8_{\pm0.1}$ | $67.4_{\pm0.1}$ |
| | CLIP-ICM* | $85.2_{\pm0.4}$ | $72.3_{\pm0.3}$ | $90.2_{\pm0.3}$ | $91.9_{\pm0.3}$ | $75.6_{\pm0.2}$ | $52.1_{\pm0.2}$ | $68.2_{\pm0.4}$ | $15.9_{\pm0.3}$ | $84.4_{\pm0.3}$ | $67.8_{\pm0.3}$ |
| | CLIP-ICM† | $84.2_{\pm0.1}$ | $71.2_{\pm0.2}$ | $89.4_{\pm0.5}$ | $91.0_{\pm0.2}$ | $74.6_{\pm0.2}$ | $50.9_{\pm0.1}$ | $67.0_{\pm0.1}$ | $14.9_{\pm0.1}$ | $83.2_{\pm0.2}$ | $66.9_{\pm0.4}$ |
| | CLIP-ICM | $\mathbf{85.9}_{\pm0.2}$ | $\mathbf{73.1}_{\pm0.3}$ | $\mathbf{90.8}_{\pm0.2}$ | $\mathbf{92.6}_{\pm0.2}$ | $\mathbf{76.4}_{\pm0.2}$ | $52.8_{\pm0.4}$ | $68.9_{\pm0.3}$ | $16.5_{\pm0.3}$ | $\mathbf{85.2}_{\pm0.2}$ | $\mathbf{68.5}_{\pm0.5}$ |

solely on invariant factors achieve consistent performance across training and test environments. Identifiability analysis further guarantees the existence of a linear mapping from CLIP embeddings to an invariant subspace, which can be estimated using interventional data. If certain conditions are met, the invariant predictor achieves a lower OOD risk. Building on these insights, we propose CLIP-ICM, a framework for collecting interventional data and estimating the required linear mapping. Experiments on benchmark datasets demonstrate the superior performance of CLIP-ICM compared to existing methods.

## Impact Statement

This paper presents work whose goal is to advance the field of Machine Learning. There are many potential societal consequences of our work, none of which we feel must be specifically highlighted here.

## Acknowledgements

The authors would like to thank the anonymous reviewers for their valuable comments. This work is supported in part by the China Postdoctoral Science Foundation, Grant No.2024M753356, and in part by the National Natural Science Foundation of China, No. 62406313.

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

# Appendix

This supplementary material provides detailed proofs for the theorem and proposition mentioned in the main text. Furthermore, additional experimental results and implementation details are provided.

- Appendix A provides the definition of all notations of the main text.

- Appendix B presents a detailed discussion of various fine-tuning strategies within the context established in Section 5.1.

- Appendix C provides some background of the causality.

- Appendix D provides the proof of Proposition 5.1 in the main text.

- Appendix E provides the proof of Proposition 5.3 in the main text.

- Appendix F provides the proof of Proposition 5.5 in the main text.

- Appendix G provides the proof of Theorem 5.6 in the main text.

- Appendix H provides the implementation details for CLIP-ICM.

- Appendix I provides the performance of CLIP-ICM in the setting of Section 4.

- Appendix J provides the experimental results in ImageNet and variants of ImageNet.

- Appendix K provides detailed description of the used datasets.

- Appendix L provides a detailed comparison with the existing works.

- Appendix M provides the ablation study.

- Appendix N provides an analysis where the domain prompt is used in zero-shot case.

- Appendix O provides full results in each domain in DomainNet Benchmark.

## A. List of Notations

We list the definitions of all notations from the main text as follows:

- **Variables**
    - $x$: An image sample.
    - $t$: A text description.
    - $e$: An environment.
    - $e^*$: The test environment.
    - $c$: A class.
    - $y$: A ground-truth label.
    - $\hat{y}$: A predicted label.
    - $\hat{z}$: The image embedding.
    - $z$: A latent factor.
    - $z_{inv}$: An invariant latent factor.
    - $z_{var}$: A variant latent factor.
    - $w$: The normal vector of the decision boundary.

- **Random Variables**
    - $X$: The random variable of the image.
    - $Y$: The random variable of the class label.
    - $Z_{inv}$: The random variable of the invariant factors.

- $Z_{var}$: The random variable of the variant factors.
- $Z_{var}^*$: The random variable of variant factors in environment $e^*$.
- $E$: The selection variable of environments.

- **Sample Spaces / Supports**

  - $\mathcal{T}$: The support of text description.
  - $\mathcal{X}$: The support of image samples.
  - $\mathcal{X}^{do(z_{inv})}$: The support of image samples when the latent factor $z_{inv}$ takes a fixed value.
  - $\mathcal{Y}$: The support of the class label.
  - $\hat{\mathcal{Z}}$: The support of the embedding.
  - $\mathcal{Z}$: The support of the latent factors.
  - $\mathcal{Z}_{inv}$: The invariant subspace.
  - $\mathcal{Z}_{var}$: The variant subspace.

- **Functions**

  - $f_I$: The image encoder.
  - $f_T$: The text encoder.
  - $f_{I^*}$: The ideal image encoder.
  - $f_{T^*}$: The ideal text encoder.
  - $g$: The ideal data-generating process.
  - $f$: The ideal representation learning process.
  - $h$: The regular predictor.
  - $h_{inv}$: The invariant predictor.
  - $\mathbb{I}$: The indicator function.
  - $W$: The coefficient matrix of a linear classifier.
  - $P$: The probability distribution.
  - $P^*$: The probability distribution in environment $e^*$.
  - $P^{do(z_{inv})}$: The distribution of other latent factors when $z_{inv}$ takes a fixed value.
  - $A_{inv}$: The mapping from $\hat{\mathcal{Z}}$ to $\mathcal{Z}_{inv}$.
  - $A_{var}$: The mapping from $\hat{\mathcal{Z}}$ to $\mathcal{Z}_{var}$.
  - $\alpha$: The data augmentation function.
  - $\beta$: The function to alter text description.

- **Sets**

  - $E_{tr}$: The set of all training environments.
  - $E_{all}$: The set of all environments.
  - $\mathcal{D}$: The datasets from training environments.
  - $\mathcal{D}^e$: The dataset in environment $e$.
  - $\mathcal{H}$: The hypothesis class of predictor.
  - $\hat{Z}$: The collection of embedding vectors.
  - $\hat{Z}^{do(z_{inv})}$: The collection of embeddings obtained from interventional data, where the invariant factors $Z_{inv}$ are fixed while the variant factors $Z_{var}$ vary.

- **Functionals**

  - $R^{\text{OOD}}$: The OOD risk.
  - $R^e$: The generalization risk in environment $e$.
  - $R_{tr}$: The generalization risk on $E_{tr}$.
  - $R^*$: The generalization risk in environment $e^*$.

- **Constants**
    - $D$: The dimension of embedding space / latent factors.
    - $D_{inv}$: The dimension of the invariant factors.
    - $D_{var}$: The dimension of the variant factors.
    - $N^e$: The number of instances in environment $e$.
    - $C$: The total number of classes.
    - $K$: The total number of data augmentations.

# B. Discussion on Fine-tune Strategies

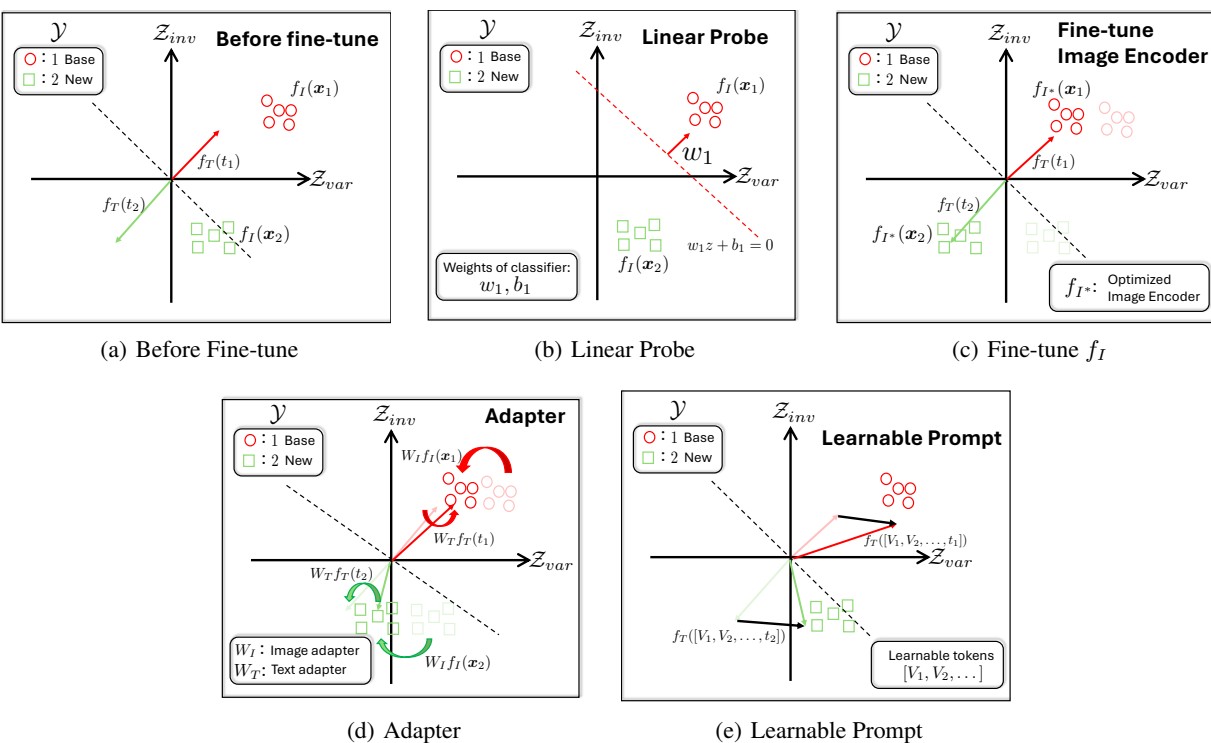

*Figure 4.* An illustration of the fine-tuning process under in the setting of Section 5.1. The red circle represents samples from the base class while the green square represents the samples from the new class.

In this section, we discuss why almost all fine-tuning methods can be understood as leading to the scenario illustrated in Figure 2. This is detailed in Figure 4.

Before fine-tuning on the training domains, the image embeddings and text embeddings are not well aligned, as shown in Figure 4(a). However, there exists class-specific text embeddings $f_T(t_1), f_T(t_2)$ for both base and new classes.

When fine-tuning with linear probe, the method learns a classifier specifically for the base classes, with parameters $w_1, b_1$. Within the base classes, the classifier parameters $w_1$ align with the image embeddings, as illustrated in Figure 4(b). Thus, it satisfies the conditions depicted in Figure 2(a). However, linear probe fine-tuning does not utilize information from CLIP's text encoder, making it unable to predict new classes.

When only the image encoder is fine-tuned, the distribution of image embeddings changes, while the text embeddings remain fixed. This adjustment aligns the image embeddings for base classes in the training domains with their corresponding text embeddings. For new classes, the image embedding distribution also shifts, and **ideally**, they should align with the fixed text embeddings, as shown in Figure 4(c). This satisfies the conditions described in Figure 2(a).

When using an adapter for fine-tuning on the training domains, both image and text embeddings are altered. This ensures that the image and text embeddings for base classes are aligned. **Ideally**, by applying the same adapter to new classes, the

text and image embeddings of new classes should also align in the training environment, as depicted in Figure 4(d).

When using learnable prompts for fine-tuning, the image embeddings remain unchanged, but the introduction of learnable tokens modifies the text embeddings. This allows the text embeddings of base classes to align with the image embeddings in the training domains. **Ideally**, by applying the same learnable tokens to new classes, the text embeddings of new classes adjust accordingly, aligning with the image embeddings in the training environment, as shown in Figure 4(e).

It is important to note that these fine-tuning methods utilize the same text description embeddings or linear classifiers during testing across environments. However, in the test environments, the distribution of image data shifts, potentially leading to the situation illustrated in Figure 2(b), where incorrect predictions occur for both base and new classes.

Worse still, when fine-tuning is performed only on the base classes, the learnable parameters may overfit to the base classes. This overfitting can result in the misalignment of embeddings in Figure 4(c), Figure 4(d), and Figure 4(e) even within the training environment for new classes. Consequently, the misalignment is further exacerbated in the test environments, making it even more challenging to correctly classify new classes.

## C. Background in Causality

**Structural Causal Model**    The SCMs can be used to describe the causal variables and their relationships. The definition of SCM is:

**Definition C.1.** (Structural Causal Model (Pearl, 2009))
1. A set $U$ of background or exogenous variables, representing factors outside the model, which nevertheless affect relationships within the model.
2. A set $V = \{V_1, \ldots, V_n\}$ of endogenous variables, assumed to be observable. Each of these variables is functionally dependent on some subset $PA_i$ of $U \cup V$. Here, $PA_i$ means the set of parent nodes of the endogenous variable $V_i$.
3. A set $F$ of functions $\{f_1, \ldots, f_n\}$ such that each $f_i$ determines the value $v_i$ of $V_i \in V$, $v_i = f_i(pa_i, u)$. These functions are also known as **causal mechanisms**.
4. A joint probability distribution $P(u)$ over $U$.

The SCMs can be represented by Directed Acyclic Graphs (DAG) (Glymour et al., 2016). In the DAG, the arrow $X \to Y$ denotes that changes in the value of $X$ directly cause changes in $Y$. However, this relationship does not hold in the reverse order.

**The Selection Diagram and Transportability**    Pearl et al. (Pearl & Bareinboim, 2014) propose the selection diagram, which is used to model the causal relation between different environments with an external selection variable. The definition of the selection diagram is:

**Definition C.2.** (Selection diagram): Let $< M, M^* >$ be a pair of SCMs relative to domain $< \Pi, \Pi^* >$, sharing a causal diagram $G$. $< M, M^* >$ is said to induce a selection diagram $D$ if $D$ is constructed as follows:
1. Every edge in $G$ is also an edge in $D$
2. $D$ contains an extra edge $S_i \to V_i$ whenever there might exist a discrepancy $f_i \neq f_i^*$ or $P(u_i) \neq P^*(u_i)$ between $M$ and $M^*$.
The set $S$ are denoted as selection variables.

Transportability aims to study whether the causal relation in a specific environment can be applied to any other environment, which is defined as:

**Definition C.3.** (Transportability): Let $D$ be a selection diagram relative to domain $< \Pi, \Pi^* >$. Let $< P, I >$ be the pair of observational and interventional distributions of $\Pi$, and $P^*$ be the observational distribution of $\Pi^*$. The causal relation $R(\Pi^*) = P^*(y|do(x))$ is said to be transportable from $\Pi$ to $\Pi^*$ in $D$ if $R(\Pi^*)$ is uniquely computable from $P, P^*, I$ in any model that induces $D$.

## D. Proof for Proposition 5.1

Before we proceed to the proof of Proposition 5.1, we first provide some useful definitions and lemma.

**Definition D.1.** ($d$-separation): A set $S$ of nodes is said to block a path $p$ if either
1. $p$ contains at least one arrow-emitting node that is in $S$,

2. $p$ contains at least one collision node that is outside $S$ and has no descendant in $S$.
If $S$ blocks all paths from set $X$ to set $Y$, it is said to $d$-separate $X$ and $Y$. $X$ and $Y$ are independent given $S$, written $X \perp\!\!\!\perp Y|S$.

The $d$-separation reflects the conditional independencies that hold the distribution $P$ that is compatible with the DAG.

**Definition D.2.** (Rules of do-calculus) Let $X, Y, Z, W$ be arbitrary disjoint sets of nodes in a causal DAG $G$. We denote by $G_{\overline{X}}$ the graph obtained by deleting from $G$ all arrows pointing to nodes in $X$. Likewise, we denote by $G_{\underline{X}}$ the graph obtained by deleting from $G$ all arrows emerging from nodes in $X$. The notation $G_{\overline{X}\underline{Z}}$ represents the deletion of both incoming and outgoing arrows.
1. Insertion/deletion of observations.

$$P(y|do(x), z, w) = P(y|do(x), w) \quad \text{if } (Y \perp\!\!\!\perp Z|X, W)_{G_{\overline{X}}}.$$

2. Action/observation exchange.

$$P(y|do(x), do(z), w) = P(y|do(x), w) \quad \text{if } (Y \perp\!\!\!\perp Z|X, W)_{G_{\overline{X}\underline{Z}}}.$$

3. Insertion/deletion of actions.

$$P(y|do(x), do(z), w) = P(y|do(x), w) \quad \text{if } (Y \perp\!\!\!\perp Z|X, W)_{G_{\overline{X}\overline{Z(W)}}}.$$

where $Z(W)$ is the set of $Z$-nodes that are not ancestors of any $W$-node in $G_{\overline{X}}$.

**Definition D.3.** (Trivial transportability). A causal relation $R$ is said to be trivially transportable from $\Pi$ to $\Pi^*$, if $R(\Pi^*)$ is identifiable from $(G^*, P^*)$.

$R$ is trivially transportable if we can directly estimate $R(\Pi^*)$ from observational data of $\Pi^*$, unaided by the causal information from $\Pi$. The following state the sufficient and necessary conditions of transportability of average causal effect $P^*(y|do(x))$.

**Definition D.4.** ($S$-admissibility). A set $Z$ of variables satisfying $(Y \perp\!\!\!\perp S|Z, X)$ in $D_{\overline{X}}$ will be called S-admissible with respect to the causal effect of $X$ on $Y$. $D_{\overline{X}}$ denote deleting all arrows pointing to node $X$ in $D$.

**Lemma D.5.** *(sufficient and necessary conditions of transportability ([Pearl & Bareinboim, 2014](#))) The average causal effect $P^*(y|do(x))$ is transportable from $\Pi$ to $\Pi^*$ if either one of the following conditions holds:*
*1. $P^*(y|do(x))$ is trivially transportable.*
*2. There exists a set of covariates $Z$ (possibly affected by $X$) such that $Z$ is $S$-admissible and for which $P^*(z|do(x))$ is transportable.*
*3. There exists a set of covariates, $W$ that satisfy $(X \perp\!\!\!\perp Y|W)_{\overline{X(W)}}$ and for which $P^*(w|do(x))$ is transportable.*

*Proof.* 1. According to Definition D.3, the causal relationship can be directly estimated from observational data of $\Pi^*$.
2. If condition 2 holds, it implies:

$$\begin{aligned} P^*(y|do(x)) &= P(y|do(x), s) \\ &= \sum_z P(y|do(x), z, s)P(z|do(x), s) \\ &= \sum_z P(y|do(x), z)P^*(z|do(x)) \end{aligned} \tag{10}$$

The transportability of $P(z|do(x))$ reduces $P^*(z|do(x))$ to a star-free expression and therefore $P^*(y|do(x))$ is transportable.
3. If condition 3 holds, it implies:

$$\begin{aligned} P^*(y|do(x)) &= P(y|do(x), s) \\ &= \sum_w P(y|do(x), w, s)P(w|do(x), s) \\ &= \sum_w P(y|w, s)P^*(w|do(x)) \end{aligned} \tag{11}$$

According to Rule 3 of Definition D.2, the transportability of $P^*(w|do(x))$ would render $P(w|do(x), s)$ to a star-free expression and render $P^*(y|do(x))$ transportable. This ends the proof. $\square$

We are now prepared to present the proof for Proposition 5.1 in the main text:

**Proposition D.6.** *Let $P^*(\cdot) := P(\cdot|e^*)$ denote the distribution in test environment $E = e^*$. The causal mechanisms are formalized through interventional distributions. We focus on the predictions $P(y|do(\cdot))$, where $do(\cdot)$ represents an intervention. The causal mechanism satisfies $P^*(y|do(\boldsymbol{x})) \neq P(y|do(\boldsymbol{x}))$. When using only $Z_{inv}$, the causal mechanism remains unchanged: $P^*(y|do(\boldsymbol{z}_{inv})) = P(y|do(\boldsymbol{z}_{inv}))$.*

*Proof.* The causal mechanism $P(y|do(\boldsymbol{x}))$ can be decomposed as:

$$P(y|do(\boldsymbol{x})) = \sum_{\boldsymbol{z}_{inv}} \sum_{\boldsymbol{z}_{var}} P(y|do(\boldsymbol{x}), \boldsymbol{z}_{var}, \boldsymbol{z}_{inv})$$
$$P(\boldsymbol{z}_{var}|do(\boldsymbol{x}))P(\boldsymbol{z}_{inv}|do(\boldsymbol{x}))$$

Similarly, the causal mechanism $P^*(y|do(\boldsymbol{x}))$ can be decomposed as:

$$P^*(y|do(\boldsymbol{x})) = P(y|do(\boldsymbol{x}), e^*) \tag{12}$$
$$= \sum_{\boldsymbol{z}_{inv}} \sum_{\boldsymbol{z}_{var}} P(y|do(\boldsymbol{x}), \boldsymbol{z}_{var}, \boldsymbol{z}_{inv}, e^*)$$
$$P(\boldsymbol{z}_{var}|do(\boldsymbol{x}), e^*)P(\boldsymbol{z}_{inv}|do(\boldsymbol{x}), e^*)$$

From the Rule 1 of Definition D.2, we have

$$P(y|do(\boldsymbol{x}), \boldsymbol{z}_{var}, \boldsymbol{z}_{inv}, e^*) = P(y|do(\boldsymbol{x}), \boldsymbol{z}_{var}, \boldsymbol{z}_{inv}), \tag{13}$$

because $Z_{var}$ satisfies $(Y \perp\!\!\!\perp E|Z_{var}, X)$ in $D_{\overline{X}}$, and according to Definition D.4, the $Z_{var}$ $E$-admissible with respect to the causal effect of $X$ on $Y$. Then, since $(Y \perp\!\!\!\perp X|Z_{var}, Z_{inv})$, the following holds:

$$P(y|do(\boldsymbol{x}), \boldsymbol{z}_{var}, \boldsymbol{z}_{inv}) = P(y|\boldsymbol{z}_{var}, \boldsymbol{z}_{inv}) \tag{14}$$

According to Definition D.1, $Z_{inv} \to Y \leftarrow Z_{var} \leftarrow E$ is a collision node:

$$P(\boldsymbol{z}_{inv}|do(\boldsymbol{x}), e^*) = P(\boldsymbol{z}_{inv}|do(\boldsymbol{x})). \tag{15}$$

And since there is no other path from $X \to Z_{inv}$ and $X \to Z_{var}$, the *do-* operator is trivial. Therefore, the Equation (12) can be rewrite as:

$$P^*(y|do(\boldsymbol{x})) = \sum_{\boldsymbol{z}_{inv}} \sum_{\boldsymbol{z}_{inv}} P(y|\boldsymbol{z}_{var}, \boldsymbol{z}_{inv})P(\boldsymbol{z}_{inv}|\boldsymbol{x})P^*(\boldsymbol{z}_{var}|\boldsymbol{x}) \tag{16}$$

Since $P^*(\boldsymbol{z}_{var}|\boldsymbol{x}) \neq P(\boldsymbol{z}_{var}|\boldsymbol{x})$, $P^*(y|do(\boldsymbol{x})) \neq P(y|do(\boldsymbol{x}))$.

If the prediction is solely based on $Z_{inv}$, $Y$ is independent of $Z_{var}$. The prediction process is now $P^*(y|do(\boldsymbol{z}_{inv})) = P(y|do(\boldsymbol{z}_{inv}), e^*)$. Since the $do(\cdot)$ cut-off all edges entering $Z_{inv}$, $Z_{inv}$ is independent of $E$. Therefore, $P^*(y|do(\boldsymbol{z}_{inv})) = P(y|do(\boldsymbol{z}_{inv}))$. This ends the proof.

$\square$

# E. Proof for Proposition 5.3

Building upon prior works (Roeder et al., 2021; Ahuja et al., 2022; Hyvarinen & Morioka, 2016; Zimmermann et al., 2021), we offer proof demonstrating that CLIP identifies latent factors up to an invertible linear transformation, and we also reuse some of their proof techniques.

**Proposition E.1.** *For an image encoder $f_I : \mathcal{X} \to \hat{\mathcal{Z}}$ and its corresponding text encoder $f_T : \mathcal{T} \to \hat{\mathcal{Z}}$, where $\hat{\mathcal{Z}} \subseteq \mathbb{R}^D$ is the embedding space. If Condition 5.2 is satisfied, then $f_I(\boldsymbol{x})$ identifies $\mathcal{Z}$ up to an invertible linear transformation $A$, i.e. $f_I(\boldsymbol{x}) = Ag^{-1}(\boldsymbol{x}) = A\boldsymbol{z}$.*

*Proof.* Consider a training dataset $\mathcal{D} = \{(x_i, t_i)\}_{i=1}^N$ sampled from the joint distribution $p(x, t)$. Let $\mathcal{T}$ denote the set of all possible values of $t$. Let $\theta$ denote the parameters of $f_I$ and $f_T$, and let $\theta^*$ denote the parameters of $f_{I^*}$ and $f_{T^*}$ (to which we have no access). The ground-truth conditional probability can be regarded as produced by $f_{I^*}$ and $f_{T^*}$:

$$p_{\theta^*}(t \mid x, \mathcal{T}) = \frac{\exp(f_{I^*}(x)^\top f_{T^*}(t))}{\sum_{t' \in \mathcal{T}} \exp(f_{I^*}(x)^\top f_{T^*}(t'))}. \tag{17}$$

Similarly, the CLIP model functions $f_I$ and $f_T$ produce the distribution:

$$p_\theta(t \mid x, \mathcal{T}) = \frac{\exp(f_I(x)^\top f_T(t))}{\sum_{t' \in \mathcal{T}} \exp(f_I(x)^\top f_T(t'))}. \tag{18}$$

The training objective for CLIP is to minimize the KL divergence $\mathbf{KL}\Big(p_\theta(t \mid x, \mathcal{T}) \,\|\, p_{\theta^*}(t \mid x, \mathcal{T})\Big)$. Ideally, after training, we have $p_\theta(t \mid x, \mathcal{T}) = p_{\theta^*}(t \mid x, \mathcal{T})$, that is:

$$\frac{\exp(f_I(x)^\top f_T(t))}{\sum_{t' \in \mathcal{T}} \exp(f_I(x)^\top f_T(t'))} = \frac{\exp(f_{I^*}(x)^\top f_{T^*}(t))}{\sum_{t' \in \mathcal{T}} \exp(f_{I^*}(x)^\top f_{T^*}(t'))}. \tag{19}$$

The above equality illustrates the consistency aspect of Condition 5.2. Building on this, for any pair $t_a$ and $t_b$ the following ratio should hold:

$$\frac{p_\theta(t_a \mid x, \mathcal{T})}{p_\theta(t_b \mid x, \mathcal{T})} = \frac{p_{\theta^*}(t_a \mid x, \mathcal{T})}{p_{\theta^*}(t_b \mid x, \mathcal{T})}. \tag{20}$$

This implies that under Condition 5.2 there exist $D + 1$ distinct text description pairs $(t_a, t_b)$ satisfying:

$$\frac{\exp(f_I(\boldsymbol{x})^T f_T(\boldsymbol{t}_a))}{\exp(f_I(\boldsymbol{x})^T f_T(\boldsymbol{t}_b))} = \frac{\exp(f_{I^*}(\boldsymbol{x})^T f_{T^*}(\boldsymbol{t}_a))}{\exp(f_{I^*}(\boldsymbol{x})^T f_{T^*}(\boldsymbol{t}_b))}. \tag{21}$$

Taking the logarithm of both sides, this simplifies to:

$$(f_T(\boldsymbol{t}_a) - f_T(\boldsymbol{t}_b))^T f_I(\boldsymbol{x}) = (f_{T^*}(\boldsymbol{t}_a) - f_{T^*}(\boldsymbol{t}_b))^T f_{I^*}(\boldsymbol{x}). \tag{22}$$

On the left-hand side of Equation (22), $(f_T(\boldsymbol{t}_a) - f_T(\boldsymbol{t}_b))$ can be treated as a basis vector. Since there exist $D + 1$ pairs of $(\boldsymbol{t}_a, \boldsymbol{t}_b)$, at least $D$ linearly independent vectors $(f_T(\boldsymbol{t}_a) - f_T(\boldsymbol{t}_b))$ can form an invertible matrix $L \in \mathbb{R}^{D \times D}$. The same holds for the right hand side, where $(f_{T^*}(\boldsymbol{t}_a) - f_{T^*}(\boldsymbol{t}_b))$ forms another matrix $L'$. Substituting these into the equation gives the following system of $D$ linear equations:

$$f_I(\boldsymbol{x}) = (L' L^{-1})^T f_{I^*}(\boldsymbol{x}). \tag{23}$$

Defining $A = L' L^{-1}$, which is invertible, we arrive at:

$$f_I(\boldsymbol{x}) = A f_{I^*}(\boldsymbol{x}) = A\boldsymbol{z} \tag{24}$$

This completes the proof. $\qquad\square$

## F. Proof for Proposition 5.5

The proof technique of Proposition 5.5 follows the approach of Ahuja et al. (2023), which we adapt to our setting.

**Proposition F.1.** *For the encoder $f_I$ satisfies Proposition 5.3, if Condition 5.4 is satisfied and sample from $P^{do(z_{inv})}(\boldsymbol{x})$ is possible. Then for any pair $(\boldsymbol{x}_1^{do(z_{inv})}, \boldsymbol{x}_2^{do(z_{inv})})$ sampled from $P^{do(z_{inv})}(\boldsymbol{x})$. The mapping $A_{inv}$ satisfies:*

$$A_{inv}(f_I(\boldsymbol{x}_1^{do(z_{inv})}) - f_I(\boldsymbol{x}_2^{do(z_{inv})})) = 0 \tag{25}$$

*Proof.* Condition 5.4 states that sampling from $P^{do(z_{inv})}(\boldsymbol{x})$ is available. The invariant factors of the images from $P^{do(z_{inv})}(\boldsymbol{x})$ are fixed at a specific value, i.e., $z_{inv} = z^\dagger$. For a sample $\boldsymbol{x}_1^{do(z_{inv})}$, its ground-truth latent representation, $\boldsymbol{z}_1^{do(z_{inv})} = g^{-1}(\boldsymbol{x}_1^{do(z_{inv})})$, can be written as $\boldsymbol{z}_1^{do(z_{inv})} = [z^\dagger, \boldsymbol{z}_{var,1}]^T$. Similarly, for another sample $\boldsymbol{x}_2^{do(z_{inv})}$, we have $\boldsymbol{z}_2^{do(z_{inv})} = [z^\dagger, \boldsymbol{z}_{var,2}]^T$.

From Proposition 5.3, the following relationships hold:

$$\begin{aligned} f_I(\boldsymbol{x}_1^{do(z_{inv})}) &= A\boldsymbol{z}_1^{do(z_{inv})}, \\ A^{-1} f_I(\boldsymbol{x}_1^{do(z_{inv})}) &= \boldsymbol{z}_1^{do(z_{inv})}, \\ \begin{bmatrix} A_{inv} \\ A_{var} \end{bmatrix} f_I(\boldsymbol{x}_1^{do(z_{inv})}) &= \begin{bmatrix} z^\dagger \\ \boldsymbol{z}_{var,1} \end{bmatrix} \end{aligned} \tag{26}$$

The same holds for $x_2^{do(z_{inv})}$. By focusing only on $A_{inv}$, we have:

$$A_{inv} f_I(x_1^{do(z_{inv})}) = z^\dagger, \tag{27}$$

and

$$A_{inv} f_I(x_2^{do(z_{inv})}) = z^\dagger, \tag{28}$$

Subtracting these two equations yields:

$$A_{inv}(f_I(x_1^{do(z_{inv})}) - f_I(x_2^{do(z_{inv})})) = 0 \tag{29}$$

This ends the proof. □

## G. Proof for Theorem 5.6

Before we proceed to the proof of Theorem 5.6, we give the following definition. Let $e_{tr} \in E_{all}$ denote the training environment. Define the distribution on some test environments $e_{te} \in E_{all}, e_{tr} \neq e_{te}$ as $P_{te}$. Let $R^e(h, h') = \mathbb{E}_{z \sim P(z)}(\ell(h(z), h'(z)))$ be the expected disagreement between two hypotheses $h, h' \in \mathcal{H}$, where $\ell$ is some loss function (e.g. cross-entropy loss). This represents a measure of how much two hypotheses disagree with each other on the training distribution. We use $\mathcal{H}\Delta\mathcal{H}$-divergence (Ben-David et al., 2010; Chuang et al., 2020) to measure whether there is any pair of hypotheses whose risk differs significantly between $P_{tr}$ and $P_{te}$.

**Definition G.1.** ($\mathcal{H}\Delta\mathcal{H}$-divergence (Ben-David et al., 2010)) Given two distribution $P_{tr}$ and $P_{te}$, and a hypothesis class $\mathcal{H}$, the $\mathcal{H}\Delta\mathcal{H}$-divergence between $P_{tr}$ and $P_{te}$ is:

$$d_{\mathcal{H}\Delta\mathcal{H}}(P_{tr}, P_{te}) := \sup_{h, h' \in \mathcal{H}} |R^{e_{tr}}(h, h') - R^{e_{te}}(h, h')|. \tag{30}$$

Next, we are going to show how the risk in $e_{tr}$ can be related to a test environment $e_{te} \in E_{all}$ with the $\mathcal{H}\Delta\mathcal{H}$-divergence.

**Lemma G.2.** *(Ben-David et al., 2010 (Ben-David et al., 2010)) For all hypothesis $h \in \mathcal{H}$, the risk on $P_{te}$ is bounded as:*

$$R^{e_{te}}(h) \leq R^{e_{tr}}(h) + d_{\mathcal{H}\Delta\mathcal{H}}(P_{tr}, P_{te}) + \lambda_{\mathcal{H}}, \tag{31}$$

*where $\lambda_{\mathcal{H}}$ is the best joint risk:*

$$\lambda_{\mathcal{H}} := \inf_{h' \in \mathcal{H}} [R^{e_{tr}}(h') + R^{e_{te}}(h')]/2. \tag{32}$$

*Proof.* By the definition of $d_{\mathcal{H}\Delta\mathcal{H}}(P_{tr}, P_{te})$,

$$d_{\mathcal{H}\Delta\mathcal{H}}(P_{tr}, P_{te}) = \sup_{h, h' \in \mathcal{H}} |R^{e_{tr}}(h, h') - R^{e_{te}}(h, h')| \tag{33}$$

Also, with the triangle inequality for classification error:

$$\begin{aligned}
R^{e_{te}}(h) &\leq R^{e_{te}}(h') + R^{e_{te}}(h, h') \\
&\leq R^{e_{te}}(h') + R^{e_{tr}}(h, h') + |R^{e_{tr}}(h, h') - R^{e_{te}}(h, h')| \\
&\leq R^{e_{te}}(h') + R^{e_{tr}}(h, h') + d_{\mathcal{H}\Delta\mathcal{H}}(P_{tr}, P_{te}) \\
&\leq R^{e_{te}}(h') + R^{e_{tr}}(h) + R^{e_{tr}}(h') + d_{\mathcal{H}\Delta\mathcal{H}}(P_{tr}, P_{te}) \\
&\leq R^{e_{tr}}(h) + d_{\mathcal{H}\Delta\mathcal{H}}(P_{tr}, P_{te}) + \lambda_{\mathcal{H}}.
\end{aligned} \tag{34}$$

This ends the proof. □

Next, we formalize some basic assumptions:

**Assumption G.3.** 1. $R^{e_{tr}}(h_{inv})$ is constant for $e \in E_{all}$.

2. The Bayes-optimal predictor $h^* \in \mathcal{H}$ and $h_{inv}^* \in \mathcal{H}$ exist.

3. The loss $\ell : \mathcal{Y} \times \mathcal{Y} \to \mathbb{R}^+$ is $L$-Lipschitz and bounded by $M > 0$.

Now we can proof Theorem 5.6 from the main text.

**Theorem G.4.** *Let $\mathcal{H}$ be a hypothesis class over $\mathcal{X} \times \mathcal{Y}$, and let $E_{all}$ denote a set of environments with distributions $\{P^e\}_{e \in E_{all}}$ over $\mathcal{X} \times \mathcal{Y}$. Let $h_{inv} \in \mathcal{H}$ be a predictor relying solely on $Z_{inv}$, and $h \in \mathcal{H}$ a predictor utilizing both $Z_{inv}$ and $Z_{var}$. If the mutual information between $Z_{inv}$ and $Z$ satisfies:*

$$I(Z_{inv}; Z) > c \quad \text{for some constant } c > 0. \tag{35}$$

*Then, the worst-case OOD risk strictly satisfies:*

$$R^{\text{OOD}}(h_{inv}) \; < \; R^{\text{OOD}}(h), \tag{36}$$

*Proof.* Step 1: Controlling In-Environment Risk via $I(Z_{inv}; Z)$

The mutual information condition $I(Z_{inv}; Z) > c$ guarantees that $Z_{inv}$ captures most predictive information in $Z$. By the data processing inequality,

$$I(Y; Z_{inv}) \geq I(Y; Z) - \epsilon(c) \tag{37}$$

where $\epsilon(c) \to 0$ as $c \to \infty$. This implies the conditional entropy gap between $h_{inv}^*$ and $h^*$ is bounded:

$$H(Y \mid Z_{inv}) - H(Y \mid Z) = I(Y; Z_{var} \mid Z_{inv}) \leq \epsilon(c). \tag{1}$$

For the $L$-Lipschitz loss, the risk gap becomes:

$$R^{e_{tr}}(h_{inv}^*) \leq R^{e_{tr}}(h^*) + L \cdot \epsilon(c). \tag{38}$$

By realizability, $h_{inv}$ approximates $h_{inv}^*$, so:

$$R^{e_{tr}}(h_{inv}) \leq R^{e_{tr}}(h_{inv}^*) + \eta \leq R^{e_{tr}}(h^*) + L \cdot \epsilon(c) + \eta, \tag{39}$$

or small $\eta > 0$.

Step 2: Bounding OOD Risk via $\mathcal{H}\Delta\mathcal{H}$-Divergence

By Definition G.1, the $\mathcal{H}\Delta\mathcal{H}$-divergence between $P_e$ and $P_{e^*}$ is:

$$d_{\mathcal{H}\Delta\mathcal{H}}(P_e, P_{e^*}) = \sup_{h, h' \in \mathcal{H}} |R^{e_{tr}}(h, h') - R^{e_{te}}(h, h')|. \tag{40}$$

For $h_{inv}$, invariance ensures minimal divergence:

$$d_{\mathcal{H}\Delta\mathcal{H}}(P_e, P_{e^*}; h_{inv}) \leq \delta_{inv}, \tag{41}$$

while for $h$, the divergence scales with $Z_{var}$-shift:

$$d_{\mathcal{H}\Delta\mathcal{H}}(P_e, P_{e^*}; h) \geq \delta_{var} \gg \delta_{inv}. \tag{42}$$

Step 3: Worst-Case Risk Comparison

By definition, $R^{\text{OOD}}(h) = \max\{R^e(h), R^{e_{te}}(h)\}$. From (2) and (3):

$$R^{e_{te}}(h) \leq R^e(h) + \delta_{var} + \lambda_{\mathcal{H}} \leq \left(R^{\text{OOD}}(h_{inv}) - L \cdot \epsilon(c) - \eta\right) + \delta_{var} + \lambda_{\mathcal{H}}. \tag{43}$$

Since the risk of $h_{inv}$ is stable ($R^{\text{OOD}}(h_{inv}) = R^e(h_{inv})$), we have:

$$R^{\text{OOD}}(h) \geq R^{e_{te}}(h) \geq R^{\text{OOD}}(h_{inv}) - L \cdot \epsilon(c) - \eta + \delta_{var} + \lambda_{\mathcal{H}}. \tag{44}$$

Under $I(Z_{inv}; Z) > c$, $\epsilon(c)$ becomes negligible. With $\delta_{var} > 0$ and $\lambda_{\mathcal{H}} > 0$, it follows that:

$$R^{\text{OOD}}(h_{inv}) < R^{\text{OOD}}(h). \tag{45}$$

This ends the proof. □

# H. Implementation of CLIP-ICM

## H.1. Interventional Data Collection

In Section 6, we propose two approaches for collecting interventional data. In the following sections, we provide a detailed explanation of each method.

**Image-based Interventional Data Collection** To generate interventional data, we employ the following data augmentation techniques: ColorJitter, GrayScale, GaussianBlur, RandomInvert, RandomRotation, RandomPosterize, RandomSolarize, and RandomEqualize. After applying these methods, the changes in the latent factors of the corresponding images are summarized in Table 4. In practice, for each image, we randomly select one augmentation technique from the set and generate an augmented sample, which is then paired with the original sample.

*Table 4.* Impact of data augmentation on specific latent factors ($Z_{inv}$: ✓, $Z_{var}$: ×)

| Augmentation Technique | Shape | Structure | Color | Texture |
|---|---|---|---|---|
| ColorJitter | ✓ | ✓ | × | × |
| GrayScale | ✓ | ✓ | × | ✓ |
| GaussianBlur | ✓ | ✓ | ✓ | × |
| RandomInvert | ✓ | ✓ | × | ✓ |
| RandomPosterize | ✓ | ✓ | × | × |
| RandomSolarize | ✓ | ✓ | × | × |
| RandomEqualize | ✓ | ✓ | × | ✓ |
| RandomRotation | × | ✓ | ✓ | ✓ |

**Text-based Interventional Data Collection** To collect text-based interventional data, two components are required: a image description model and a text intervention model. We utilize GPT-4o (OpenAI, 2023) for both purposes. The text description model takes an image as input, processes it using a carefully crafted prompt, and outputs a descriptive caption of the image. The text intervention model then takes this caption as input, applies a specified intervention using a modified prompt, and outputs an altered version of the original description.

---

Prompt for image description : Describe this image in detail.

---

Prompt for text intervention :

You are tasked with creating interventional descriptions of images for a classification task. Given the following JSON input:

''' {

"image_description": "A detailed textual description of the image, including all features (relevant and irrelevant to the classification task).",

"class_labels": ["Class1", "Class2", "Class3"]

} '''

Your goal is to:

1. Identify features related to the classification task that are invariant to environmental shifts. These features should remain unchanged in the output description.

2. Intervene on all other features (e.g., texture, background, lighting) by replacing their description with new, plausible alternatives.

3. Maintain the original structure of the description while altering only the specified features.

---

Output a single JSON object containing only the '"intervened_description"', which reflects these modifications. "'

We provide examples of both descriptions for samples from the PACS dataset in Figure 5.

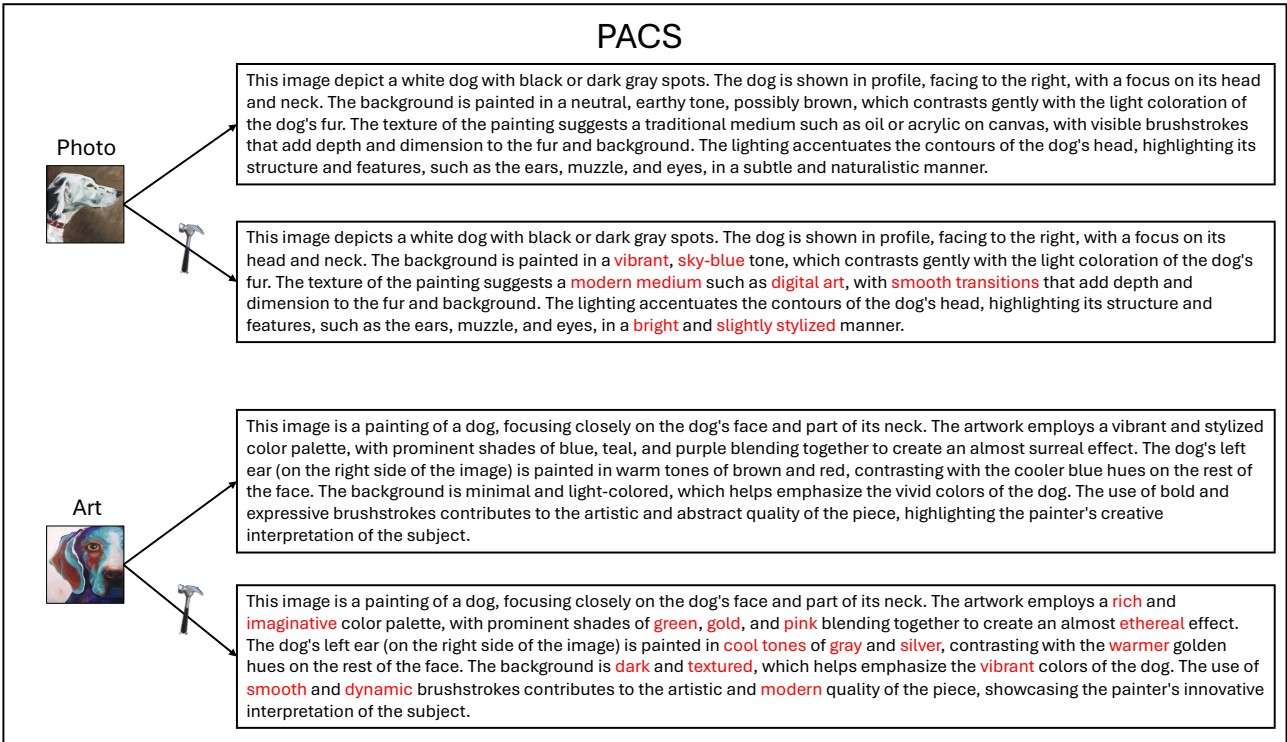

*Figure 5.* The example images with image description and the interventional description from the PACS dataset.

### H.2. Details in Estimating $A_{inv}$

The process begins with collecting $\hat{Z}$ and $\hat{Z}^{do(z_{inv})}$. For any given image sample $\boldsymbol{x}$, we first generate an augmented version $\alpha(\boldsymbol{x})$ using data augmentation. We then obtain the corresponding text description $\boldsymbol{t}$ using a text description model and create an intervened description $\beta(\boldsymbol{t})$ using a text intervention model. This allows us to construct four embedding pairs: $(f_T(\boldsymbol{t}), f_T(\boldsymbol{t}) - f_T(\beta(\boldsymbol{t})))$, $(f_T(\boldsymbol{t}), f_T(\boldsymbol{t}) - f_I(\alpha(\boldsymbol{x})))$, $(f_I(\boldsymbol{x}), f_I(\boldsymbol{x}) - f_I(\alpha(\boldsymbol{x})))$, and $(f_I(\boldsymbol{x}), f_I(\boldsymbol{x}) - f_T(\beta(\boldsymbol{t})))$. These pairs are stacked across all samples to form $\hat{Z}$ and $\hat{Z} - \hat{Z}^{do(z_{inv})}$.

Once the data is collected, the next step is to fit $A_{inv}$. One approach is to use principal component analysis (PCA) to compute $A_{inv}$ that satisfies Equation (9). Another approach is to treat $A_{inv}$ as a learnable matrix and optimize it directly using gradient descent.

### H.3. Pseudo Code

The pseodo code of CLIP-ICM is illustrated in Algorithm 1.

### H.4. Computation Cost and Data Collection Cost

All experiments are conducted on a single NVIDIA-RTX A6000 GPU, and the average training duration for each result is less than one hour.

**Computation Complexity Analysis**    We provide a detailed analysis of the computational complexity of CLIP-ICM.

---

**Algorithm 1** CLIP-ICM

---

**Require:** CLIP image encoder $f_I$, text encoder $f_T$ and images from training domain $\boldsymbol{x} \in \mathcal{D}$, $\mathcal{D} = \{\mathcal{D}^e\}_{e \in E_{tr}}$. $K$ types of data
augmentation $\mathcal{A} = \{\alpha_k\}_{k=1}^K$. Hyperparameter $\lambda$ and $D_{inv}$. Text description of all images $\boldsymbol{x} \in \mathcal{D}$. The text intervention model $\beta$.

 1: Initialize the container $\hat{Z} = [\quad]$ for original representation
 2: Initialize the container $\hat{Z}' = [\quad]$ for intervened representation
 3: **for** $\boldsymbol{x}$ in $\mathcal{D}$ **do**
 4:     Sample random augmentation $\alpha_k$ from $K$
 5:     Get the text description $\boldsymbol{t}$ of $\boldsymbol{x}$.
 6:     Get the original image representation $\hat{z} = f_I(\boldsymbol{x})$
 7:     Get the original text representation $\hat{z}_t = f_T(\boldsymbol{t})$
 8:     Get $\boldsymbol{x}^{do(z_{inv})} = \alpha_k(\boldsymbol{x})$
 9:     Get $\boldsymbol{t}^{do(z_{inv})} = \beta(\boldsymbol{t})$
10:     Get the intervened representation $\hat{z}' = f_I(\alpha_k(\boldsymbol{x}))$
11:     Get the intervened representation $\hat{z}'_t = f_T(\beta(\boldsymbol{t}))$
12:     Append $\hat{z}$ into $\hat{Z}$ // image pair
13:     Append $\hat{z}'$ into $\hat{Z}'$
14:     Append $\hat{z}_t$ into $\hat{Z}$ // text pair
15:     Append $\hat{z}'_t$ into $\hat{Z}'$
16:     Append $\hat{z}$ into $\hat{Z}$ // image-text pair
17:     Append $\hat{z}'_t$ into $\hat{Z}'$
18:     Append $\hat{z}_t$ into $\hat{Z}$ // text-image pair
19:     Append $\hat{z}'$ into $\hat{Z}'$
20: **end for**
21: Get the correlation matrix $Corr = \hat{Z}^T\hat{Z} - \lambda \cdot (\hat{Z} - \hat{Z}')^T(\hat{Z} - \hat{Z}')$. $\hat{Z} \in \mathbb{R}^{N \times d}, \hat{Z}' \in \mathbb{R}^{N \times d}, Corr \in \mathbb{R}^{d \times d}$
22: Perform Eigenvalue Decomposition for $Corr$ to get $U, V, \Sigma$
23: Get the highest $D_{inv}$ singular vector $U_{:D_{inv},:} \in \mathbb{R}^{D_{inv} \times D}$
**Output:** The target transform $A_{inv}$ is $U_{:D_{inv},:}$

---

According to Algorithm 1, the training process of CLIP-ICM can be divided into four key steps: data preprocessing, obtaining the representation of the training data, calculating $Corr$, and obtaining the projection matrix through SVD. Let $N$ denote the size of the training dataset and $D$ the dimension of the representation:

1. **Data Preprocessing**: Since data augmentation is applied to individual samples, the computational complexity for this step is $O(N)$.

2. **Obtaining Representations**: The pre-trained network is applied to individual samples. Like the previous step, this operation also has a complexity of $O(N)$.

3. **Calculating the** $Corr$: Let $Z \in \mathbb{R}^{N \times D}$ be the matrix containing all training data representations. The complexity of computing $Z^T Z$ is $O(D^2 N)$. Since $D$ is the dimension of representation and is independent of the dataset size, it can be considered constant, the complexity is also $O(N)$.

4. **SVD of the Correlation Matrix**: The computational complexity of SVD for the correlation matrix is $O(D^3)$. However, since the matrix $Corr \in \mathbb{R}^{D \times D}$ is independent of the dataset size, this step's complexity can be considered as $O(1)$.

In summary, CLIP-ICM has a computational complexity of $O(N)$, growing linearly with dataset size. For larger datasets, $A_{inv}$ is treated as a learnable matrix and optimized via gradient descent, requiring only a linear matrix in $\mathbb{R}^{D \times D_{inv}}$. In both approaches, CLIP-ICM remains highly efficient.

**Data Collection Cost** We use GPT-4o as the generator for text-based interventional data, which involves three steps: processing image inputs, generating textual descriptions, and producing intervention-modified descriptions. For smaller datasets such as PACS, VLCS, Terra Incognita, and Office-Home, we generate interventional text data for all image samples. For large datasets such as DomainNet and ImageNet, we sample 20% of the dataset for text-based interventions. Data is collected using the GPT-4o batch API at a cost of \$1.25 per million input tokens and \$5 per million output tokens. The total cost for data collection is \$162.79.

# I. OOD Performance in Terra Incognita

We evaluate CLIP-ICM using the same experimental setup as described in Section 4, with results presented in Table 5. The results show that CLIP-ICM Linear-Probe improves performance by an average of 6.3% compared to direct Linear-Probe. Furthermore, CLIP-ICM significantly outperforms naive fine-Tune in new classes under both open-class and domain shift conditions.

*Table 5.* The performance of CLIP on the Terra Incognita dataset in accuracy (%). L100, L38, L43, and L46 represent different environments. In the Linear probe case, the values in ($\cdot$) represent the accuracy achieved by directly fine-tuning with the linear probe on the target domain.

| | DOMAIN SHIFT | | | | |
|---|---|---|---|---|---|
| METHOD | L100 | L38 | L43 | L46 | AVG |
| LINEAR-PROBE | 73.6 (90.9) | 58.3 (86.3) | 61.0 (76.0) | 47.8 (78.9) | 60.2 (83.0) |
| CLIP-ICM LINEAR-PROBE | 79.8 (↑ 6.2) | 61.5 (↑ 3.2) | 66.9 (↑ 5.9) | 57.9 (↑ 10.1) | 66.5 (↑ 6.3) |

| | OPEN CLASS UNDER DOMAIN SHIFT | | | | | |
|---|---|---|---|---|---|---|
| SPLIT | METHOD | L100 | L38 | L43 | L46 | AVG |
| BASE | ZERO-SHOT | 56.4 | 23.2 | 30.4 | 25.8 | 33.9 |
| | FINE-TUNE | 75.6 | 58.5 | 65.2 | 55.1 | 63.6 |
| | CLIP-ICM | 63.7 | 52.6 | 58.8 | 51.2 | 56.6 |
| NEW | ZERO-SHOT | 47.6 | 17.6 | 35.2 | 37.4 | 34.5 |
| | FINE-TUNE | 36.7 | 11.2 | 25.1 | 25.5 | 24.6 |
| | CLIP-ICM | 59.4 | 43.2 | 51.6 | 46.8 | 50.2 |
| TOTAL | ZERO-SHOT | 52.0 | 20.4 | 32.8 | 31.6 | 34.2 |
| | FINE-TUNE | 56.2 | 34.9 | 45.2 | 40.3 | 44.2 |
| | CLIP-ICM | 61.6 | 47.9 | 55.2 | 49.0 | 53.4 |

# J. More Results

We also utilize ImageNet as the training data and validated the OOD generalization ability of CLIP-ICM on four datasets: ImageNet V2 (Recht et al., 2019), ImageNet-A (Hendrycks et al., 2021b), and ImageNet-Sketch (Wang et al., 2019), ImageNet R (Hendrycks et al., 2021a). Specifically:

- ImageNet V2 (Recht et al., 2019) contains new test data for the ImageNet benchmark, providing a fresh evaluation of model performance.

- ImageNet-A (Hendrycks et al., 2021b) consists of real-world, unmodified, and naturally occurring examples that are often misclassified by ResNet models, presenting challenging cases for OOD generalization.

- ImageNet-Sketch (Wang et al., 2019) includes sketch representations of ImageNet classes, offering a test of the model's ability to generalize to different visual styles.

- ImageNet-R (Hendrycks et al., 2021a) contains art, cartoons, deviantart, graffiti, embroidery, graphics, origami, paintings, patterns, plastic objects, plush objects, sculptures, sketches, tattoos, toys, and video game renditions of ImageNet classes.

We additionally conducted experiments on the iWildCam-WILDS 2020 dataset (Koh et al., 2021). iWildCAM comprises 203,029 images from 182 different animal species, which were collected from 323 camera traps distributed across various locations. The images obtained from different locations exhibit variations in lighting, color, camera angle, background, vegetation, and relative animal frequencies. We follow the setting in (Koh et al., 2021), use images from 243 locations as the training domain, and those from 48 other locations as the test domain. We report the average macro F1 score of CLIP, CLIP-ICM*, CLIP-ICM†, and CLIP-ICM under both ID and OOD conditions, as shown in the Table 7: From the results in Table 7, we can observe that regardless of type of interventional data, CLIP-ICM consistently outperforms CLIP in both ID and OOD settings.

Table 6. Accuracy on ImageNet with various domain shifts.

| METHOD | IN-DISTRIBUTION | OUT-OF-DISTRIBUTIONS | | | | |
|---|---|---|---|---|---|---|
| | IMAGENET | IMAGENET-V2 | IMAGENET-S | IMAGENET-A | IMAGENET-R | AVG. |
| ZERO-SHOT | 66.7 | 60.8 | 46.1 | 47.8 | 74.0 | 57.2 |
| FINE-TUNE | $68.2_{\pm 0.1}$ | $61.9_{\pm 0.1}$ | $46.8_{\pm 0.1}$ | $46.4_{\pm 0.1}$ | $75.1_{\pm 0.1}$ | $57.6_{\pm 0.1}$ |
| COOP (ZHOU ET AL., 2022B) | 71.5 | 64.2 | 48.0 | 49.7 | 75.2 | 59.3 |
| COCOOP (ZHOU ET AL., 2022A) | 71.0 | 64.2 | 48.8 | 50.6 | 76.2 | 59.9 |
| CLIPOOD (SHU ET AL., 2023) | $\mathbf{71.6}_{\pm 0.1}$ | $64.9_{\pm 0.1}$ | $49.3_{\pm 0.1}$ | $50.4_{\pm 0.1}$ | $77.2_{\pm 0.1}$ | $60.4_{\pm 0.1}$ |
| CLIP-ICM* | $70.5_{\pm 0.3}$ | $64.7_{\pm 0.5}$ | $49.9_{\pm 0.4}$ | $50.3_{\pm 0.3}$ | $76.2_{\pm 0.2}$ | $60.2_{\pm 0.4}$ |
| CLIP-ICM† | $71.2_{\pm 0.4}$ | $63.7_{\pm 0.3}$ | $48.8_{\pm 0.4}$ | $46.8_{\pm 0.1}$ | $74.8_{\pm 0.2}$ | $58.5_{\pm 0.3}$ |
| CLIP-ICM | $\mathbf{71.6}_{\pm 0.1}$ | $\mathbf{65.8}_{\pm 0.3}$ | $\mathbf{50.9}_{\pm 0.4}$ | $\mathbf{51.4}_{\pm 0.3}$ | $\mathbf{77.6}_{\pm 0.2}$ | $\mathbf{61.4}_{\pm 0.3}$ |

Table 7. Accuracy on the iWildCam-WILDS 2020 dataset.

| Method | ID (48 Locations) | OOD (243 Locations) |
|---|---|---|
| CLIP | 14.2 | 10.6 |
| CLIP-ICM* | 15.6 | 13.3 |
| CLIP-ICM† | 15.2 | 12.2 |
| CLIP-ICM | **15.8** | **14.1** |
| CLIP Linear-Probe | 54.6 | 41.4 |
| CLIP-ICM* Linear-Probe | 56.2 | 42.2 |
| CLIP-ICM† Linear-Probe | 55.6 | 44.3 |
| CLIP-ICM Linear-Probe | **57.1** | **46.1** |

# K. Dataset Details from DomainBed Benchmark

DOMAINBED includes downloaders and loaders for seven multi-domain image classification tasks:

- **PACS** (Li et al., 2017) comprises four domains $e \in \{$art, cartoons, photos, sketches$\}$. This dataset contains 9,991 examples of dimension (3, 224, 224) and 7 classes.

- **VLCS** (Fang et al., 2013) comprises photographic domains $e \in \{$Caltech101, LabelMe, SUN09, VOC2007$\}$. This dataset contains 10,729 examples of dimension (3, 224, 224) and 5 classes.

- **Office-Home** (Venkateswara et al., 2017) includes domains $e \in \{$art, clipart, product, real$\}$. This dataset contains 15,588 examples of dimension (3, 224, 224) and 65 classes.

- **Terra Incognita** (Beery et al., 2018) contains photographs of wild animals taken by camera traps at locations $e \in \{$L100, L38, L43, L46$\}$. Our version of this dataset contains 24,788 examples of dimension (3, 224, 224) and 10 classes.

- **DomainNet** (Peng et al., 2019) has six domains $e \in \{$clipart, infograph, painting, quickdraw, real, sketch$\}$. This dataset contains 586,575 examples of size (3, 224, 224) and 345 classes.

For all datasets, we first pool the raw training, validation, and testing images together. For each random seed, we then instantiate random training, validation, and testing splits.

# L. More comparison with related works

## L.1. Invariant Risk Minimization (Arjovsky et al., 2020)

Invariant Risk Minimization (IRM) and IRM-based methods (Ahuja et al., 2021; Yang et al., 2023) also aim to minimize the OOD risks with an invariant predictor. However, note that these methods obtain invariant predictors without identifying the invariant factors. Moreover, as suggested in Figure 2 of Arjovsky et al. 2020 (Arjovsky et al., 2020), there are multiple solutions to the IRM objective. Therefore finding an IRM solution doesn't mean the identification of $Z_{inv}$. Compared to these methods, our method ensures the identification of $Z_{inv}$.

**L.2. Model-based Domain Generalization (Robey et al., 2021)**

Model-based Domain Generalization (Robey et al., 2021) also also proposes an SCM, their analysis method is different from ours. They believe that in domain shift, an instance $X^e$ is generated by an underlying random variable $X$ and $e$ jointly passing through a domain-transfer model $G(X, e)$. In our causal diagram analysis, we believe that the sample is generated by $Z_{var}$ and $Z_{inv}$ jointly.

MBDG obtains the invariant features by utilizing a pre-trained domain-transfer model to constrain the distance between the features extracted by the feature extractor and the representations generated by the domain-transfer model.

In contrast, our approach learns a linear transformation of the pre-trained representation to obtain features that are invariant to data augmentation. We provide the comparison results between our method and MBDG in Table 15 and Table 12.

**L.3. Subspace Identification Guarantee (Li et al., 2024)**

From the perspective of SCM analysis, SIG views the data generation process through 4 kinds of latent factors. In our SCM, $Z_{inv}$ can be regarded as a domain-invariant variable, and $Z_{var}$ can be regarded as a domain-variant variable. However, in our causal diagram, we do not distinguish between label-irrelevant and label-relevant variables in our modeling.

From a methodological perspective, SIG uses an end-to-end, reconstruction-based network, while we use a pre-trained CLIP backbone network. SIG uses a variation-inference method to identify latent factors, while we use an intervention method to identify latent factors. We present the performance of our method alongside the SIG to the Office-Home dataset in Table 13.

**L.4. Domain Prompt Learning (Zhang et al., 2023b)**

(Zhang et al., 2023b) introduces Domain Prompt Learning (DPL), a novel approach that improves the domain generalization performance of the CLIP model on various datasets by generating conditional prompts, achieving higher accuracy without fine-tuning the entire model. DPL can be regarded as a Test-Time Adaptation (TTA) approach, while our method can be regarded as a domain generalization approach. Nevertheless, we provide the comparison results in Table 2.

**L.5. Self-Supervised Learning with Data Augmentations Provably Isolates Content from Style (Von Kügelgen et al., 2021)**

(Von Kügelgen et al., 2021) also interpret data augmentations as counterfactuals, and use them to obtain the invariant factors. Our work differs from theirs in three key aspects: the background setting, the research problem, and the implementation method.

Firstly, the problem studied by (Von Kügelgen et al., 2021) is set within the framework of self-supervised contrastive learning. In their paper, the term "invariant factor" refers to the invariant parts derived from two different augmented perspectives within a positive sample pair. In contrast, the "invariant factor" in our paper refers to the latent factors that remain unchanged across different domains. Due to the diversity of augmentation methods employed in self-supervision (Chen et al., 2020), the invariant factors they investigate do not align with the invariant factors we examine.

Second, our theoretical approach diverges fundamentally from (Von Kügelgen et al., 2021) in both methodology and objectives. While their work establishes that **non-linear** encoders can achieve block identification of content factors through embedding space alignment of view-augmented samples, our analysis proceeds through two critical theoretical advancements: (1) We first demonstrate that CLIP learns representations that identify all latent factors up to an invertible linear transformation; (2) We subsequently prove that interventional data enables estimation of a **linear mapping** from the embedding space to invariant subspaces.

Finally, the research objectives of the two works are fundamentally distinct. While (Von Kügelgen et al., 2021) focuses on analyzing SimCLR's factor identification capabilities, our work specifically addresses CLIP's out-of-distribution generalization limitations by developing linear mappings to invariant subspaces. This difference in both theoretical focus and practical application underscores the novel contributions of our approach.

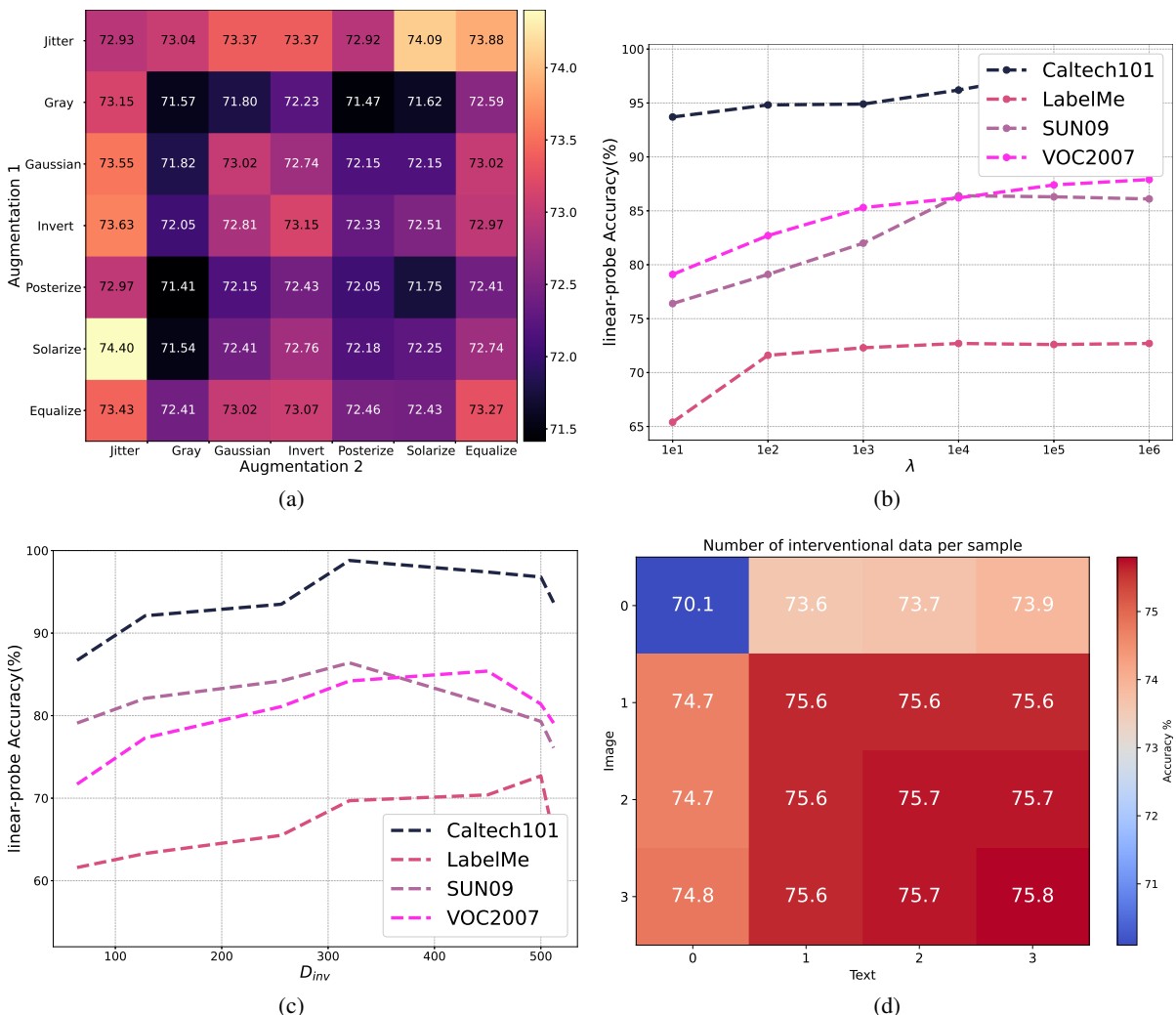

*Figure 6.* The experimental results for ablation studies. (a) The zero-shot accuracy of CLIP-ICM on the LabelMe domain of the VLCS dataset with different combinations of augmentations, and brighter colors indicate a higher accuracy. (b) The linear-probe accuracies of CLIP-ICM on the VLCS dataset with different $\lambda$. (c) The accuracy of CLIP-ICM on VLCS with different choice of $D_{inv}$. (d) The accuracy of CLIP-ICM on the LabelMe domain of the VLCS dataset with different numbers of interventional data pairs.

## M. Ablation Study

### M.1. The impact of different data augmentation

We investigate how CLIP-ICM performs with varying data augmentation strategies on the VLCS dataset. In this experiment, only image-based interventional data are utilized. The experiments include different combinations of augmentation techniques, and the prediction accuracy on the LabelMe domain is illustrated in Figure 6(a). The diagonal of Figure 6(a) indicates scenarios where only one augmentation is applied, while other cells represent combinations of data augmentation techniques. It is noteworthy that the use of data augmentation significantly improves performance, with certain combinations showing even more promising results.

### M.2. The impact of hyperparameter $\lambda$.

We conduct experiments on the VLCS dataset and visualize the results in Figure 6(b). In this experiment, only image-based interventional data are utilized. From the results, we can observe that $\lambda$ has a significant impact on the prediction results. This is because that higher value of $\lambda$ indicates the second term of Equation (9) is more important and has a higher impact.

These results illustrate the effectiveness of our proposed method.

### M.3. Ablation study on dimension of $Z_{inv}$

As we mentioned at Section 6, the dimension $D_{inv}$ of $Z_{inv}$ is a hyperparameter. To investigate the influence of $D_{inv}$, we conduct an ablation study regarding the choice of $D_{inv}$. All results are the performance CLIP-ICM with only image-based interventional data utilized, conducted on the VLCS dataset with ViT-B/16 as the backbone, where $D_{inv} < 512$. The results are shown in Figure 6(c).

From Figure 6(c) we can see that as the $D_{inv}$ increases, the accuracy first increases, stabilizes, slowly decreases, and finally drops. Therefore, the optimum dimension of $D_{inv}$ should be around 300 to 350 for all domains in VLCS.

### M.4. Number of Interventional data pairs.

By default, each sample from the training environments is paired with an image-based interventional sample via data augmentation and a text-based interventional sample generated by an LLM. In this section, we examine whether increasing interventional data enhances CLIP-ICM's performance, as shown in Figure 6(d). The results indicate that while adding more data pairs slightly improves performance, the gain is marginal.

### M.5. The role of $A_{inv}$

To validate the role of $A_{inv}$, we conduct an ablation study to illustrate its contribution. Specifically, we use our generated image-based interventional data to train a linear probe on the DomainBed benchmark. The experimental results are presented in Table 8. From the results, we observe that incorporating image-based interventional data into the linear probe leads to

*Table 8.* Ablation results on the DomainBed benchmark under domain shift. We compare the standard linear probe, linear probe trained with image-based interventional data, and CLIP-ICM* linear probe trained with the proposed $A_{inv}$ module. Methods with * indicate training with only image-based interventional data.

| Method | PACS | VLCS | OfficeHome | TerraInc | DomainNet | AVG. |
|---|---|---|---|---|---|---|
| Linear-Probe | 96.4 | 78.7 | 81.9 | 60.2 | 55.0 | 74.4 |
| Linear-Probe + Interventional data | 96.8 | 79.3 | 82.3 | 60.5 | 55.8 | 74.9 |
| CLIP-ICM* Linear-Probe | **97.5** | **86.5** | **84.6** | **64.3** | **64.0** | **79.0** |

only marginal improvements. In contrast, although both settings utilize the same interventional data, the performance of CLIP* + Linear-Probe is significantly higher. These findings demonstrate that our proposed $A_{inv}$ module substantially enhances the OOD generalization ability of CLIP.

## N. An analysis of the zero-shot prediction with domain prompt

In experiments from Section 7, we use the standard text prompt template when performing zero-shot prediction, i.e. "A photo of [cls]" where [cls] is the name of the corresponding class.

To explore the influence of text prompts on zero-shot prediction, we conducted experiments using the PACS, VLCS, and Office-Home datasets. In these experiments, we employed text prompts that incorporate domain information. For datasets where the domain names hold particular significance, such as PACS and Office-Home, we utilized the prompt 'A [domain] of [cls]'. In contrast, for the VLCS dataset, where the domain names do not have specific meanings, we adopted the prompt 'A photo of [cls] in [domain]'. Here, [domain] represents the name of the test domain. The results are illustrated in Table 9, Table 10 and Table 11.

It is important that the objective of domain generalization is to evaluate the performance of a model on unseen domains. Therefore, incorporating domain information, such as (a sketch of [cls]) would not align with the requirements of the domain generalization task. We only conduct this experiment for exploration.

From the results in Table 9, we find that only in some domains, incorporating domain information in the prompt template can improve the zero-shot performance, such as Sketch in PACS, and Clipart in Office-Home. However, in most domains, where the domain name doesn't provide any information, or can't describe the domain properly, the performance is lower

than the original template.

Another important observation is that CLIP-ICM consistently improves the performance of CLIP in most domains, without any domain-related information or other kinds of template design.

*Table 9.* Comparison results of zero-shot performance of CLIP with domain prompt in PACS dataset. ↑ denotes that the results of CLIP with domain prompt are higher than standard prompt, while ↓ denotes that the results of CLIP with domain prompt are lower than standard prompt.

| Algorithm | Photo | Art | Cartoon | Sketch | Avg |
|---|---|---|---|---|---|
| CLIP | **100.0** | 97.4 | 99.2 | 88.1 | 96.1 |
| CLIP + Domain Prompt | **100.0** − | 97.4 − | 99.0 ↓ | 90.2 ↑ | 96.6 ↑ |
| CLIP-ICM | 99.1 | **99.5** | **99.7** | **92.2** | **97.7** |

*Table 10.* Comparison results of zero-shot performance of CLIP with domain prompt in VLCS dataset. ↑ denotes that the results of CLIP with domain prompt are higher than standard prompt, while ↓ denotes that the results of CLIP with domain prompt are lower than standard prompt.

| Algorithm | Caltech101 | LabelMe | SUN09 | VOC2007 | Avg |
|---|---|---|---|---|---|
| CLIP | 99.9 | 70.1 | 73.5 | 86.1 | 82.4 |
| CLIP + Domain Prompt | 99.2 ↓ | 69.7 ↓ | 65.1 ↓ | 80.3 ↓ | 78.5 ↓ |
| CLIP-ICM | **100.0** | **75.6** | **79.2** | **90.0** | **86.2** |

*Table 11.* Comparison results of zero-shot performance of CLIP with domain prompt in Office-Home dataset. ↑ denotes that the results of CLIP with domain prompt are higher than standard prompt, while ↓ denotes that the results of CLIP with domain prompt are lower than standard prompt.

| Algorithm | Art | Clipart | Product | Real | Avg |
|---|---|---|---|---|---|
| CLIP | 83.3 | 65.3 | 89.0 | 89.3 | 81.7 |
| CLIP + Domain Prompt | 82.4 ↓ | 67.0 ↑ | 87.8 ↓ | 88.7 ↓ | 81.4 ↓ |
| CLIP-ICM | **84.6** | **70.7** | **91.7** | **91.5** | **84.6** |

## O. Full Results on Domainbed

In this section, we provide the full results on datasets from the Domainbed benchmark.

*Table 12.* Accuracy on PACS dataset. **A**,**C**,**P** and **S** represents different domains.

| Algorithm | P | A | C | S | Avg |
|---|---|---|---|---|---|
| Backbone is ResNet-50 | | | | | |
| ERM | 97.2±0.3 | 84.7±0.4 | 80.8±0.6 | 79.3±1.0 | 85.5 |
| IRM (Arjovsky et al., 2020) | 96.7±0.6 | 84.8±1.3 | 76.4±1.1 | 76.1±1.0 | 83.5 |
| GroupDRO (Sagawa et al.) | 96.7±0.3 | 83.5±0.9 | 79.1±0.6 | 78.3±2.0 | 84.4 |
| Mixup (Yan et al., 2020) | 97.6±0.1 | 86.1±0.5 | 78.9±0.8 | 75.8±1.8 | 84.6 |
| MMD (Li et al., 2018) | 96.6±0.2 | 86.1±1.4 | 79.4±0.9 | 76.5±0.5 | 84.6 |
| DANN (Ganin et al., 2016) | 97.3±0.4 | 86.4±0.8 | 77.4±0.8 | 73.5±2.3 | 83.6 |
| ARM (Zhang et al., 2021) | 97.4±0.3 | 86.8±0.6 | 76.8±0.5 | 79.3±1.2 | 85.1 |
| MBDG (Robey et al., 2021) | 97.0 | 80.6 | 79.3 | **85.2** | 85.6 |
| CLIP Linear-probe | 99.2±0.7 | 91.7±0.8 | 92.5±0.7 | 82.3±0.2 | 91.4 |
| CLIP Zero-shot | 99.3±0.0 | 91.0±0.0 | 93.1±0.0 | 80.5±0.0 | 91.0 |
| CLIP-ICM* | 99.4±0.4 | 93.6±0.2 | 99.3±0.5 | 83.1±0.6 | 93.8 |
| CLIP-ICM* Linear-probe | 99.4±0.5 | 92.8±0.7 | 93.9±0.4 | 83.8±0.5 | 92.5 |
| CLIP-ICM† | 99.0±0.2 | 93.3±0.4 | 98.9±0.1 | 82.8±0.4 | 93.8 |
| CLIP-ICM† Linear-probe | 99.1±0.5 | 92.5±0.4 | 93.4±0.3 | 83.5±0.4 | 92.5 |
| CLIP-ICM | 99.8±0.4 | **94.2±0.2** | **99.7±0.4** | 83.7±0.5 | **93.8** |
| CLIP-ICM Linear-probe | **100.0±0.0** | 93.4±0.2 | 94.5±0.4 | **84.2±0.2** | 92.5 |
| Backbone is ViT-B/16 | | | | | |
| CLIP Zero-shot | **100.0±0.0** | 97.4±0.0 | 99.2±0.0 | 88.1±0.0 | 96.1 |
| CLIP Linear-probe | 97.3±0.7 | 98.4±0.1 | 99.5±0.4 | 90.4±0.3 | 96.4 |
| CLIP-ICM* | 98.7±0.5 | 99.1±0.4 | 99.9±0.1 | 91.7±0.4 | 97.3 |
| CLIP-ICM* Linear-probe | 98.8±0.2 | 99.4±0.3 | 99.8±0.8 | 92.1±0.5 | 97.5 |
| CLIP-ICM† | 98.3±0.4 | 98.6±0.4 | 99.5±0.3 | 90.7±0.3 | 96.8 |
| CLIP-ICM† Linear-probe | 98.4±0.1 | 98.9±0.3 | 99.3±0.1 | 92.1±0.1 | 97.2 |
| CLIP-ICM | 99.1±0.4 | 99.5±0.2 | 99.7±0.3 | 92.2±0.4 | 97.7 |
| CLIP-ICM Linear-probe | 99.1±0.4 | **99.9±0.1** | **99.9±0.1** | **92.4±0.1** | **97.8** |

*Table 13.* Accuracy on Office-Home dataset. **A**,**C**,**P** and **R** represents different domains.

| Algorithm | A | C | P | R | Avg |
|---|---|---|---|---|---|
| Backbone is ResNet-50 | | | | | |
| ERM | 61.3±0.7 | 52.4±0.3 | 75.8±0.1 | 76.6±0.3 | 66.5 |
| IRM (Arjovsky et al., 2020) | 58.9±2.3 | 52.2±1.6 | 72.1±2.9 | 74.0±2.5 | 64.3 |
| GroupDRO (Sagawa et al.) | 60.4±0.7 | 52.7±1.0 | 75.0±0.7 | 76.0±0.7 | 66.0 |
| Mixup (Yan et al., 2020) | 62.4±0.8 | 54.8±0.6 | 76.9±0.3 | 78.3±0.2 | 68.1 |
| MMD (Li et al., 2018) | 60.4±0.2 | 53.3±0.3 | 74.3±0.1 | 77.4±0.6 | 66.3 |
| DANN (Ganin et al., 2016) | 59.9±1.3 | 53.0±0.3 | 73.6±0.7 | 76.9±0.5 | 65.9 |
| ARM (Zhang et al., 2021) | 58.9±0.8 | 51.0±0.5 | 74.1±0.1 | 75.2±0.3 | 64.8 |
| SIG (Li et al., 2024) | 76.4 | **63.9** | 85.4 | 85.8 | 77.8 |
| CLIP Zero-shot | 71.3±0.0 | 50.4±0.0 | 81.7±0.0 | 82.6±0.0 | 71.5 |
| CLIP Linear-probe | 68.0±0.8 | 46.3±0.3 | 80.4±0.9 | 81.9±0.7 | 69.1 |
| CLIP-ICM* | 72.6±0.4 | 55.0±0.9 | 83.2±0.3 | 83.7±0.8 | 73.6 |
| CLIP-ICM* Linear-probe | 78.3±0.3 | 56.4±0.8 | 88.6±0.8 | 87.7±0.7 | 77.8 |
| CLIP-ICM† | 72.2±0.4 | 54.6±0.5 | 82.7±0.2 | 83.3±0.3 | 73.2 |
| CLIP-ICM† Linear-probe | 77.9±0.1 | 55.9±0.4 | 88.2±0.5 | 87.3±0.2 | 77.3 |
| CLIP-ICM | 72.9±0.2 | 55.4±0.2 | 83.6±0.1 | 84.2±0.3 | 74.0 |
| CLIP-ICM Linear-probe | **78.8±0.4** | 56.7±0.3 | **89.1±0.4** | **88.1±0.2** | **78.2** |
| Backbone is ViT-B/16 | | | | | |
| CLIP Zero-shot | 83.3±0.0 | 65.3±0.0 | 89.0±0.0 | 89.3±0.0 | 81.7 |
| CLIP Linear-probe | 78.9±0.9 | 69.3±0.9 | 90.3±0.3 | 89.0±0.2 | 81.9 |
| CLIP-ICM* | 83.7±0.7 | 67.4±0.6 | 90.8±0.9 | 90.1±0.6 | 82.6 |
| CLIP-ICM* Linear-probe | 84.3±0.5 | 71.4±0.2 | 92.5±0.2 | 90.2±0.8 | 84.6 |
| CLIP-ICM† | 82.8±0.4 | 66.8±0.3 | 89.8±0.2 | 89.0±0.2 | 82.1 |
| CLIP-ICM† Linear-probe | 83.3±0.4 | 70.5±0.4 | 91.4±0.4 | 84.4±0.2 | 82.4 |
| CLIP-ICM | **84.6±0.3** | 70.7±0.3 | 91.7±0.3 | **91.5±0.4** | 84.6 |
| CLIP-ICM Linear-probe | 84.3±0.5 | **71.4±0.2** | **92.5±0.2** | 90.2±0.8 | **87.1** |

*Table 14.* Accuracy on TerraIncognita dataset.

| Algorithm | L100 | L38 | L43 | L46 | Avg |
|---|---|---|---|---|---|
| Backbone is ResNet-50 | | | | | |
| ERM | 49.8±4.4 | 42.1±1.4 | 56.9±1.8 | 35.7±3.9 | 46.1 |
| IRM (Arjovsky et al., 2020) | 54.6±1.3 | 39.8±1.9 | 56.2±1.8 | 39.6±0.8 | 47.6 |
| GroupDRO (Sagawa et al.) | 41.2±0.7 | 38.6±2.1 | 56.7±0.9 | 36.4±2.1 | 43.2 |
| Mixup (Yan et al., 2020) | 59.6±2.0 | 42.2±1.4 | 55.9±0.8 | 33.9±1.4 | 47.9 |
| MMD (Li et al., 2018) | 41.9±3.0 | 34.8±1.0 | 57.0±1.9 | 35.2±1.8 | 42.2 |
| DANN (Ganin et al., 2016) | 51.1±3.5 | 40.6±0.6 | 57.4±0.5 | 37.7±1.8 | 46.7 |
| ARM (Zhang et al., 2021) | 49.3±0.7 | 38.3±2.4 | 55.8±0.8 | 38.7±1.3 | 45.5 |
| CLIP Zero-shot | 7.7±0.0 | 14.8±0.0 | 32.4±0.0 | 20.9±0.0 | 19.0 |
| CLIP Linear-probe | 52.0±1.0 | 34.4±0.6 | 56.1±0.9 | 32.8±0.2 | 44.1 |
| CLIP-ICM* | 38.8±0.4 | 33.6±0.6 | 33.6±0.1 | 30.1±0.7 | 34.0 |
| CLIP-ICM* Linear-probe | 64.1±0.4 | 46.5±0.7 | 61.1±0.6 | 42.8±0.4 | 53.6 |
| CLIP-ICM† | 38.3±0.1 | 32.9±0.4 | 33.0±0.2 | 29.4±0.4 | 33.4 |
| CLIP-ICM† Linear-probe | 63.5±0.2 | 45.9±0.5 | 60.3±0.2 | 42.0±0.2 | 52.9 |
| CLIP-ICM | 39.6±0.3 | 34.5±0.5 | 34.8±0.4 | 31.2±0.3 | 35.0 |
| CLIP-ICM Linear-probe | **65.3±0.5** | **47.0±0.2** | **62.0±0.3** | **43.4±0.3** | **54.4** |
| Backbone is ViT-B/16 | | | | | |
| CLIPOOD (Shu et al., 2023) | 74.6±0.3 | 57.3±0.5 | 59.1±0.9 | 47.7±0.9 | 59.7 |
| CLIP Zero-shot | 52.0±0.0 | 20.4±0.0 | 32.8±0.0 | 31.6±0.0 | 34.2 |
| CLIP Linear-probe | 73.6±0.5 | 58.3±0.7 | 61.0±0.1 | 47.8±1.0 | 60.2 |
| CLIP-ICM* | 59.3±0.4 | 46.2±0.4 | 52.4±0.7 | 41.6±0.8 | 49.9 |
| CLIP-ICM* Linear-probe | 78.7±0.5 | 60.2±0.9 | 66.0±0.8 | 52.4±0.1 | 64.3 |
| CLIP-ICM† | 58.5±0.2 | 45.2±0.2 | 51.6±0.2 | 37.4±0.2 | 48.2 |
| CLIP-ICM† Linear-probe | 77.9±0.3 | 59.1±0.2 | 65.4±0.1 | 50.5±0.2 | 63.2 |
| CLIP-ICM | 60.8±0.1 | 47.3±0.4 | 53.5±0.1 | 48.4±0.1 | 52.5 |
| CLIP-ICM Linear-probe | **79.8±0.3** | **61.5±0.3** | **66.9±0.3** | **57.9±0.3** | **66.5** |

*Table 15.* Accuracy on VLCS dataset.

| Algorithm | C | L | S | V | Avg |
|---|---|---|---|---|---|
| Backbone is ResNet-50 | | | | | |
| ERM | 97.7±0.4 | 64.3±0.9 | 73.4±0.5 | 74.6±1.3 | 77.5 |
| IRM (Arjovsky et al., 2020) | 98.6±0.1 | 64.9±0.9 | 73.4±0.6 | 77.3±0.9 | 78.5 |
| GroupDRO (Sagawa et al.) | 97.3±0.3 | 63.4±0.9 | 69.5±0.8 | 76.7±0.7 | 76.7 |
| Mixup (Yan et al., 2020) | 98.3±0.6 | 64.8±1.0 | 72.1±0.5 | 74.3±0.8 | 77.4 |
| MMD (Li et al., 2018) | 97.7±0.1 | 64.0±1.1 | 72.8±0.2 | 75.3±3.3 | 77.5 |
| DANN (Ganin et al., 2016) | 99.0±0.3 | 65.1±1.4 | 73.1±0.3 | 77.2±0.6 | 78.6 |
| ARM (Zhang et al., 2021) | 98.7±0.2 | 63.6±0.7 | 71.3±1.2 | 76.7±0.6 | 77.6 |
| MBDG (Robey et al., 2021) | 98.3 | 68.1 | 68.8 | 76.3 | 77.9 |
| CLIP Zero-shot | 99.2±0.0 | 69.5±0.0 | 69.8±0.0 | 84.9±0.0 | 80.9 |
| CLIP Linear-probe | 98.1±0.7 | 63.8±0.8 | 79.8±0.7 | 84.5±1.0 | 81.5 |
| CLIP-ICM* | 99.4±0.4 | 71.8±0.8 | 71.7±0.8 | 85.3±1.0 | 82.1 |
| CLIP-ICM* Linear-probe | 99.2±0.1 | 69.9±0.3 | 80.5±0.3 | 89.5±0.9 | 84.8 |
| CLIP-ICM† | 98.7±0.5 | 71.1±0.2 | 71.0±0.4 | 84.4±0.4 | 81.3 |
| CLIP-ICM† Linear-probe | 98.4±0.3 | 69.3±0.2 | 79.8±0.2 | 88.6±0.2 | 84.0 |
| CLIP-ICM | **100.0±0.0** | **72.4±0.2** | 72.5±0.4 | 86.0±0.2 | 82.7 |
| CLIP-ICM Linear-probe | 99.7±0.3 | 70.7±0.2 | **81.0±0.4** | **90.0±0.3** | **85.4** |
| Backbone is ViT-B/16 | | | | | |
| CLIPOOD (Shu et al., 2023) | 97.5±0.6 | 68.3±0.5 | 83.9±0.9 | 88.7±1.0 | 84.6 |
| CLIP Zero-shot | 99.9±0.0 | 70.1±0.0 | 73.5±0.0 | 86.1±0.0 | 82.4 |
| CLIP Linear-probe | 93.7±0.2 | 65.4±0.2 | 76.4±0.2 | 79.1±0.3 | 78.7 |
| CLIP-ICM* | 100.0±0.0 | 74.7±0.4 | 74.5±0.4 | 87.1±0.7 | 84.1 |
| CLIP-ICM* Linear-probe | 98.8±0.3 | 72.7±0.8 | 86.4±0.9 | 87.9±0.9 | 86.5 |
| CLIP-ICM† | 98.8±0.2 | 73.6±0.2 | 73.9±0.5 | 87.3±0.4 | 83.4 |
| CLIP-ICM† Linear-probe | 98.2±0.4 | 71.7±0.4 | 85.8±0.1 | 85.1±0.2 | 85.2 |
| CLIP-ICM | **100.0±0.0** | **75.6±0.2** | 79.2±0.4 | **90.0±0.1** | 86.2 |
| CLIP-ICM Linear-probe | 99.8±0.3 | 73.9±0.3 | **87.3±0.5** | 85.5±0.3 | **86.6** |

Table 16. Accuracy on DomainNet dataset.

| Algorithm | CLIPART | INFOGRAPH | PAINTING | QUICKDRAW | REAL | SKETCH | Avg |
|---|---|---|---|---|---|---|---|
| | | | Backbone is ResNet-50 | | | | |
| ERM | 58.1±0.3 | 18.8±0.3 | 46.7±0.3 | 12.2±0.4 | 59.6±0.1 | 49.8±0.4 | 40.9 |
| IRM (Arjovsky et al., 2020) | 48.5±2.8 | 15.0±1.5 | 38.3±4.3 | 10.9±0.5 | 48.2±5.2 | 42.3±3.1 | 33.9 |
| GroupDRO (Sagawa et al.) | 47.2±0.5 | 17.5±0.4 | 33.8±0.5 | 9.3±0.3 | 51.6±0.4 | 40.1±0.6 | 33.3 |
| Mixup (Yan et al., 2020) | 55.7±0.3 | 18.5±0.5 | 44.3±0.5 | 12.5±0.4 | 55.8±0.3 | 48.2±0.5 | 39.2 |
| MMD (Li et al., 2018) | 32.1±13.3 | 11.0±4.6 | 26.8±11.3 | 8.7±2.1 | 32.7±13.8 | 28.9±11.9 | 23.4 |
| DANN (Ganin et al., 2016) | 53.1±0.2 | 18.3±0.1 | 44.2±0.7 | 11.8±0.1 | 55.5±0.4 | 46.8±0.6 | 38.3 |
| ARM (Zhang et al., 2021) | 49.7±0.3 | 16.3±0.5 | 40.9±1.1 | 9.4±0.1 | 53.4±0.4 | 43.5±0.4 | 35.5 |
| CLIP Zero-shot | 53.1±0.0 | 39.2±0.0 | 52.9±0.0 | 5.7±0.0 | 76.7±0.0 | 48.0±0.0 | 45.9 |
| CLIP Linear-probe | 59.2±0.1 | 38.6±0.9 | 54.1±0.7 | 8.2±0.7 | 56.4±0.7 | 39.6±0.7 | 42.6 |
| CLIP-ICM* | 58.0±0.1 | 42.3±0.5 | 54.4±0.2 | 12.8±0.7 | 78.2±0.1 | 49.6±0.2 | 49.2 |
| CLIP-ICM* Linear-probe | 63.1±0.1 | 50.1±0.2 | 66.2±1.0 | 19.7±0.3 | 83.8±0.4 | 63.2±0.3 | 57.8 |
| CLIP-ICM[†] | 57.4±0.2 | 41.7±0.3 | 53.9±0.1 | 12.1±0.5 | 77.6±0.3 | 49.0±0.3 | 48.6 |
| CLIP-ICM[†] Linear-probe | 62.4±0.2 | 49.4±0.2 | 65.7±0.1 | 18.9±0.4 | 83.2±0.5 | 62.6±0.2 | 57.0 |
| CLIP-ICM | 59.1±0.3 | 43.4±0.2 | 55.5±0.3 | 13.8±0.2 | 79.1±0.3 | 50.8±0.3 | 50.3 |
| CLIP-ICM Linear-probe | **64.1±0.2** | **51.3±0.5** | **67.2±0.5** | **20.6±0.5** | **84.9±0.3** | **64.1±0.2** | **58.7** |
| | | | Backbone is ViT-B/16 | | | | |
| CLIPOOD (Shu et al., 2023) | 77.6 | 54.7 | 72.5 | 20.7 | 85.7 | 69.9 | 63.5 |
| CLIP Zero-shot | 70.2±0.0 | 46.6±0.0 | 65.0±0.0 | 13.7±0.0 | 82.9±0.0 | 62.7±0.0 | 56.8 |
| CLIP Linear-probe | 73.9±0.5 | 40.2±0.5 | 62.1±0.2 | 15.1±0.3 | 76.9±0.1 | 62.0±0.7 | 55.0 |
| CLIP-ICM* | 77.1±0.5 | 51.8±0.5 | 67.9±1.0 | 16.7±0.3 | 82.9±0.8 | 66.9±0.2 | 60.5 |
| CLIP-ICM* Linear-probe | 78.6±0.2 | 55.6±0.2 | 72.9±0.6 | 22.1±0.7 | 86.1±0.3 | 68.5±0.3 | 64.0 |
| CLIP-ICM[†] | 75.9±0.5 | 50.1±0.3 | 66.2±0.4 | 15.0±0.5 | 81.4±0.1 | 55.8±0.3 | 57.4 |
| CLIP-ICM[†] Linear-probe | 76.7±0.3 | 53.5±0.2 | 70.6±0.4 | 20.8±0.4 | 84.0±0.2 | 52.1±0.5 | 59.6 |
| CLIP-ICM | 78.1±0.4 | 52.8±0.3 | 69.0±0.2 | 17.6±0.4 | 84.2±0.2 | 64.8±0.2 | 61.1 |
| CLIP-ICM Linear-probe | **79.6±0.2** | **56.7±0.2** | **74.1±0.3** | **23.5±0.2** | **87.3±0.2** | **68.9±0.3** | **65.0** |

