# OpenReview forum: "Learning Invariant Causal Mechanism from Vision-Language Models"
_ICML.cc/2025/Conference — ICML 2025 poster_

### Official Review · Reviewer_5jTK · 2025-03-10

**Overall Recommendation:** 1

**Summary:**

This work aims to leverage Invariant Causal Mechanisms in causality to improve prediction under distribution shifts. However, a detailed summary is challenging for me due to several fundamental issues, including an unclear problem formulation, misconceptions of key concepts, and unrealistic theoretical assumptions.

**Claims And Evidence:**

There are several unclear or even mistaken claims in the paper. For example:

1) Problem Setting: The claim "The goal of OOD generalization is to learn a predictor from training environments... domain shift and open-class scenarios. Domain shift arises when the data distribution in the test environment differs from that in the training environment, while open-class scenarios involve previously unseen classes appearing at test time." is confusing.

Domain shift is a broad category encompassing various settings, such as covariate shift, conditional shift, and label distribution shift. The authors should specify which type of domain shift their work addresses to avoid ambiguity.
Open-class scenarios, where previously unseen classes appear at test time, present a significant challenge. The authors should clarify whether this setting is realistically addressable and, if so, whether there exists a theoretical solution for it.

2) Conceptual Misuse: You have assumed a causal generative model, as shown on the left in Figure 1, where there is a clear causal relationship, e.g., y causes z, and z causes x. Causal mechanisms should be defined from cause to effect, rather than as p(y∣do(x)) or p(y∣do(z)), as claimed. From my understanding, a causal mechanism refers to the underlying process or system that explains how one variable influences another in a causal relationship. It describes how causes bring about effects, and is typically assumed to be invariant. Therefore, one cannot claim that the relationship from effect to cause constitutes a causal mechanism, as this relationship is generally variant and does not align with the principles of causal inference. Further, in Proposition 5.1, p(y|do(z)) or p(y|do (x)) (e.g., "cause given the effect") typically does not have a well-defined meaning in the standard framework. Please let me know if I have misunderstood this.

**Essential References Not Discussed:**

N/A

**Experimental Designs Or Analyses:**

For the experiment results based on CLIP, there is a significant concern regarding whether the training process of CLIP truly does not use the data in the experiments. Since CLIP is trained on a large number of image-text pairs, it’s important to question whether there is any potential data leakage. Specifically, it should be clarified whether the data used in the experiments overlaps with or has any connection to the data CLIP was trained on, as this could lead to biased or invalid results. Ensuring that no data leakage occurs is critical to maintaining the integrity of the experiment's findings.

**Methods And Evaluation Criteria:**

Since the identifiability theory is problematic (see below for concerns regarding the theory), I am not confident that the method's effectiveness is due to causality.

**Other Comments Or Suggestions:**

Causality is a challenging concept to understand. I believe it is particularly effective in handling distribution shift tasks, as it not only provides a theoretical framework but also offers practical tools in certain cases. However, we must be cautious in how we apply it, and at the very least, it requires a deep understanding of causality.

**Other Strengths And Weaknesses:**

N/A

**Questions For Authors:**

Overall,
1) the problem setting is unclear, and some fundamental concepts in causality are misused (see Claims And Evidence).

2) The identifiability analysis is unrealistic and nearly flawed (Theoretical Claims), which undermines confidence in the proposed methods.

3) Additionally, using the CLIP model to claim OOD distribution experiments should be approached with caution and carefully considered (see Experimental Designs Or Analyses).

**Relation To Broader Scientific Literature:**

N/A

**Theoretical Claims:**

Theorem 5.3 is central to supporting this work. However, the assumption in Condition 5.2 is quite peculiar. It broadly states, "There exist some samples such that the inference model can be equal to the generative model on these samples." This is strange, because the generative model is completely unknown. How can one enforce the inference model to match an unknown prior from the generative model? If this assumption holds, one could simply assume that the inference model equals the generative model, which would make the proof trivial. In fact, after reviewing the proof, I found that there is almost no technical challenge to the identifiability proof under the assumption in Condition 5.2.

---

> ### Author Rebuttal · Authors · 2025-03-31
>
> ## Re: Claims And Evidence & Q1
>
> **Regarding the definition of domain shift and open-class.**
>
> 1. In our context, domain shift primarily refers to covariate shift, where the $p(x)$ differs between the training and testing phases while  $p(y|x)$ remains unchanged. This scenario is widely adopted in standard domain generalization tasks [1].
>
> 2. The open-class prediction problem is well defined [2–5]. The problem is addressable,  and one of the solution is CLIP [3]. Several studies [4,5] also provide theoretical support.
>
> We will include detailed explanations regarding these scenarios in the final version.
>
> **Regarding the causal mechanism in Section 5.1.**
>
> We clarify as follows:
>
> 1. The construction of SCM depends on the **type of task** [10]. For example, whether the chicken causes the egg or the egg causes the chicken depends on the objective: if we study how chickens produce eggs, the chicken is the cause and the egg is the effect; if we study how eggs develop into chickens, then the egg is the cause. Therefore, when we study the task of prediction, the input image is the cause, and the predicted label is the effect.
> 2. We study **two** SCMs in our paper: Figure 1(a) is the generation process, while Figure 1(b) is the prediction process. Since our work primarily focuses on prediction, the causal mechanisms $p(y|do(x))$ and $p(y|do(z_{inv}))$ in Proposition 5.1 are both defined based on the SCM **in Figure 1(b)** rather than Figure 1(a). In Figure 1(b), the model construct a causal chain $X \to Z \to Y$, where $X$ is the cause, $Y$ is the effect. Therefore, it **does not means inferring the cause from the effect**. In this SCM, the causal effect is: "how change in image $X$ affects the prediction of output $Y$".
> 3. As stated in lines 194–197, the prediction process can be viewed as the inverse of the data generation process. We emphasize that the term “inverse process” here is **solely a mathematical construct** to derive the structural equations. These equations correspond to edges in Figure 1(b), which we further provide a detailed explanation in lines 197–208. Therefore, Figure 1(b) is a valid SCM.
>
> In summary, our proposed SCM doesn't contain "cause given the effect" scenarios. And the causal mechanism in Figure 1(b) is valid and well-defined.
>
> ## Re: Theoretical Claims & Q2
>
> 1. The reviewer may have misunderstood the logical connection between Condition 5.2 and Theorem 5.3. Theorem 5.3 aims to prove identifiability under certain conditions, and our work focuses on **formulating such a condition**—namely, Condition 5.2. Therefore, although the proof of Theorem 5.3 is relatively straightforward, its validity relies on the formulation of Condition 5.2.
> 2. Literature [6] proves that without additional constraint, latent factors are unidentifiable. Motivated by literatures [7,8,9], we **identify and formalize** this condition in this paper, thereby facilitating a clear understanding and straightforward implementation of the proof for Theorem 5.3.
> 3. The **advantage** of this condition is that it **does not require** additional assumptions on the latent factors (prior distribution) or on the generative process. Instead, it relies solely on observable label data.
> 4. The condition consists of two parts: consistency and diversity. The consistency (Equation 12) only requires the output distribution $\hat{p}(y|x)$ to match the **observable distribution** $p(y|x)$ , rather than an unknown prior.
> 5. We demonstrate in lines 247~255 why CLIP can be considered satisfied this condition.
>
> In summary, the Condition 5.2 and Theorem 5.3 are formulated within a standard theoretical framework based on extensive prior work and have practical implications.
>
> ## Re: Experimental Designs Or Analyses & Q3
>
> We understand the reviewer's concerns, but:
>
> 1. To date, OpenAI has not released the training data used for CLIP, which makes it extremely challenging to verify whether there is any overlap between the experimental data and the data used to train CLIP.
> 2. Our experimental design strictly adheres to the established community standards for fine-tuning CLIP in domain generalization tasks. (including CLIP-Adapter, CLIPood, CoOp, CoCoOp, MIRO, and DPL)
>
> Therefore, we believe that our experimental setup is both reasonable and widely accepted.
>
>
> [1] Domain generalization: A survey.
>
> [2] A survey of zero-shot learning: Settings, methods, and applications.
>
> [3] Learning transferable visual models from natural language supervision.
>
> [4] Zero-shot learning with semantic output codes.
>
> [5] Attribute-based classification for zero-shot visual object categorization.
>
> [6] Nonlinear independent component analysis: Existence and uniqueness results.
>
> [7] Nonlinear ICA using auxiliary variables and generalized contrastive learning.
>
> [8] On linear identifiability of learned representations.
>
> [9] Contrastive learning inverts the data generating process.
>
> [10] Toward causal representation learning.

---

> > ### Comment · Reviewer_5jTK · 2025-04-02
> >
> > **In our context, domain shift primarily refers to covariate shift**..
> >
> > --I respectfully disagree. Domain shift can generally be categorized into several specific settings, including covariate shift, target shift, conditional shift, and conditional-target shift [1,2].
> >
> > **The open-class prediction problem is well defined [2–5]. The problem is addressable, and one of the solution is CLIP [3]**
> >
> > --How do you ensure that the training data used for CLIP does not include previously unseen classes from the testing data?
> >
> >
> > **We study two SCMs in our paper:**,
> >
> > --For a given context, there should typically be only one causal model, as a causal model aims to represent a physical process. One cannot claim two models for the same context, as the corresponding physical process is determined and unique. You have defined data generation in Figure 1a. In this context, Figure 1b—which you acknowledge as a predictive model—should only be understood as an inference model."
> >
> > **Condition 5.2 and Theorem 5.3.**
> >
> > --Theorem 5.3 is based on Condition 5.2. If Condition 5.2 is not satisfied, Theorem 5.3 does not hold. From a high-level perspective, Condition 5.2 requires that the estimated z (the left-hand side of Eq. 2, $\hat{z}= f_{I}(x) = f\_{I}(g(z))$ ) matches the ground-truth $z$ (the left-hand side of Eq. 2, where $ f\_{I*}(x) = g^{-1}(x) = g^{-1}(g(z))=z$). Consequently, you assume that $\hat{z} = z$, which is the objective of identifiability.......Moreover, one does not know the ground-truth $z$. Even if one were to assume it, how, then, could this condition be incorporated into the inference model?
> >
> >
> > [1] Zhang, Kun, et al. "Domain adaptation under target and conditional shift." International conference on machine learning. Pmlr, 2013.
> >
> > [2] Stojanov, Petar, et al. "Domain adaptation with invariant representation learning: What transformations to learn?." Advances in Neural Information Processing Systems 34 (2021): 24791-24803.

---

> > > ### Author Response · Authors · 2025-04-03
> > >
> > > **Response to Comment 1:**
> > >
> > > Indeed, the understanding of domain shift is as the reviewer described. However, what we intended to express is that our submission **focuses specifically** on covariate shift, that is, the discrepancy between the training and testing data distributions.
> > >
> > > **Response to Comment 2:**
> > >
> > > Since the composition of CLIP's training dataset has not been publicly released, we are unable to directly verify its contents. To further investigate this issue, we propose an experimental approach. The basic idea is as follows: if the dataset used in our submission were included in CLIP’s training data, then testing CLIP directly on this dataset should yield strong performance.
> > >
> > > |     Method     | IMAGENET-S | IMAGENET-A | Terra Incognita | iWildCam-WILDS 2020 |
> > > | :------------: | :--------: | :--------: | :-------------: | :-----------------: |
> > > | CLIP Zero-shot |    46.1    |    47.8    |      34.2       |        10.6         |
> > > |      Ours      |    50.9    |    51.4    |      52.5       |        14.1         |
> > >
> > > The results of our test are shown in the table below. We observe that CLIP's performance is clearly suboptimal when tested directly. This supports our claim that the dataset is not included in CLIP's training data, and also validates the soundness of our experimental design.
> > >
> > > **Response to Comment 3:**
> > >
> > > In our previous response, we provided an example: *Which came first, the chicken or the egg?* This example was intended to illustrate the following point: while it is true that the SCM remains invariant, the true SCM is also unknown. We can only infer it based on empirical observations and reasoning. As a result, different interpretations may lead to different SCMs.
> > >
> > > In this paper, we present two such interpretations: one from the perspective of data generation, and the other from the perspective of data prediction. These two interpretations form a closed loop—they are mutually reversible. Building on this, the remainder of the paper develops the framework from the prediction-oriented perspective.
> > >
> > > **Response to Comment 4:**
> > >
> > > Condition 5.2 does not imply that $\hat{z}=z$. We provide a detailed explanation below.
> > >
> > > Consider a training dataset
> > > $$
> > > \mathcal{D} = \{(x_i, t_i)\}_{i=1}^N,
> > > $$
> > > sampled from the joint distribution $p(x,t)$. Let $\mathcal{T}$ denote the set of all possible values of $t$.
> > >
> > > Let $\theta$ denote the parameters of $f_I$ and $f_T$, and let $\theta^*$ denote the parameters of $f_{I^*}$ and $f_{T^*}$ (to which we have no access).
> > >
> > > The ground-truth conditional probability can be regarded as produced by $f_{I^*}$ and $f_{T^*}$:
> > > $$
> > > p_{\theta^*}(t\mid x,\mathcal{T}) = \frac{\exp(f_{I^*}(x)^\top f_{T^*}(t))}{\sum_{t'\in\mathcal{T}} \exp(f_{I^*}(x)^\top f_{T^*}(t'))}
> > > =\begin{cases}
> > > 1, & \text{if } (x,t)\in\mathcal{D},\\\\
> > > 0, & \text{otherwise}.
> > > \end{cases}
> > > $$
> > >
> > > Similarly, the CLIP model functions $f_I$ and $f_T$ produce the distribution
> > > $$
> > > p_{\theta}(t\mid x,\mathcal{T}) = \frac{\exp(f_{I}(x)^\top f_{T}(t))}{\sum_{t'\in\mathcal{T}} \exp(f_{I}(x)^\top f_{T}(t'))}.
> > > $$
> > >
> > > The training objective for CLIP is to minimize the KL divergence
> > > $$
> > > \mathbf{KL}\Bigl(p_{\theta}(t\mid x,\mathcal{T}) \Vert p_{\theta^*}(t\mid x,\mathcal{T})\Bigr).
> > > $$
> > > Ideally, after training, we have
> > > $$
> > > p_{\theta}(t\mid x,\mathcal{T}) = p_{\theta^*}(t\mid x,\mathcal{T}),
> > > $$
> > > that is,
> > > $$
> > > \frac{\exp(f_{I}(x)^\top f_{T}(t))}{\sum_{t'\in\mathcal{T}} \exp(f_{I}(x)^\top f_{T}(t'))}
> > > = \frac{\exp(f_{I^*}(x)^\top f_{T^*}(t))}{\sum_{t'\in\mathcal{T}} \exp(f_{I^*}(x)^\top f_{T^*}(t'))}.
> > > $$
> > >
> > > This equality illustrates the consistency aspect of Condition 5.2. Building on this, for any pair $t_a$ and $t_b$ the following ratio should hold:
> > > $$
> > > \frac{p_{\theta}(t_a\mid x,\mathcal{T})}{p_{\theta}(t_b\mid x,\mathcal{T})} = \frac{p_{\theta^*}(t_a\mid x,\mathcal{T})}{p_{\theta^*}(t_b\mid x,\mathcal{T})},
> > > $$
> > > which implies
> > > $$
> > > \frac{\exp(f_{I}(x)^\top f_{T}(t_a))}{\exp(f_{I}(x)^\top f_{T}(t_b))}
> > > = \frac{\exp(f_{I^*}(x)^\top f_{T^*}(t_a))}{\exp(f_{I^*}(x)^\top f_{T^*}(t_b))}.
> > > $$
> > > Taking logarithms on both sides, we obtain
> > > $$
> > > \bigl(f_T(t_a) - f_T(t_b)\bigr)^\top f_I(x) = \bigl(f_{T^*}(t_a) - f_{T^*}(t_b)\bigr)^\top f_{I^*}(x).
> > > $$
> > >
> > > Moreover, the diversity condition requires that there exist at least $D+1$ pairs $(t_a, t_b)$ such that different $f_T(t_a)-f_T(t_b)$ form different basis for some space $L$, and different $f_{T^*}(t_a)-f_{T^*}(t_b)$ form different basis for another space $L'$. Consequently, we have
> > > $$
> > > f_I(x) = \bigl(L' L^{-1}\bigr)^\top f_{I^*}(x) = A f_{I^*}(x),
> > > $$
> > > indicating that $f_I(x)$ is a linear transformation of $f_{I^*}(x)$. Note that the matrix $A$ is unknown.
> > >
> > > Thus, Condition 5.2 does not require any knowledge of $f_{I^*}$ or $f_{T^*}$, nor does it necessitate knowing the ground-truth $z$. Instead, we only assume that the data distribution is generated by these underlying functions.

---

### Official Review · Reviewer_HsFQ · 2025-03-12

**Overall Recommendation:** 4

**Summary:**

The paper analyzes the OOD generalization of CLIP via the lens of causal/invariant predictor learning, where the goal is to make predictions via the invariant (causal) features for the downstream task. Motivated by the failure cases of naive funetuning of CLIP, the authors propose CLIP-ICM as a principled approach. The proposed approach relies on the linear identifiability guarantees in CLIP's representation space, which is further disentangled into invariant and environment specific features by leveraging interventional data. With the identified invariant features, CLIP-ICM train a linear probe for making predictions in the downstream task. CLIP-ICM is benchmarked on widely used OOD generalization datasets, where it outperforms baselines (existing strategies for finetuning CLIP) especially for the case of open class domain shifts.


## Update after rebuttal

I have read the author rebuttal and other reviews as well. I think the paper is very interesting, technically sound, and make a good use for proposing methodology inspired from the latent identification literature. Hence, I retrain my rating and vouch for acceptance.

**Claims And Evidence:**

Yes, the claims made in the submission are well supported with clear and convincing evidence.

**Strong empirical evidence for the claims**

- The failure cases with naive finetuning strategies of CLIP are highlighted clearly with experiments on the Terra Incognita dataset (Table 1).

- The main experiments in Table 2 test CLIP-ICM with a variety of baselines on multiple benchmarks, with CLIP-ICM providing improved  performance in nearly all the cases. Further, experiments in Table 3 with results for the open classes domain shift strengthen the author's claim of superior OOD generalization.

- Given the requirement of interventional data in CLIP-ICM, the authors generate interventional data to test by manipulating both the base images and captions. This helps to analyze CLIP-ICM's performance with access to diverse interventional data, and ablations CLIP-ICM$^{\star}$ and CLIP-ICM$^{\dagger}$ provide further details.

**Essential References Not Discussed:**

No, I believe all essential references have been discussed to the best of my knowledge. The authors have written have a very detailed related works section.

**Experimental Designs Or Analyses:**

Yes, I checked the soundness/validity of all the experiments in the paper, and the experiment design doesn't have any flaws. Further, the authors have done a good job at analyzing their findings, its coherent with the experiment results.

**Methods And Evaluation Criteria:**

Yes, the proposed methods and evaluation criteria make sense for the problem at hand. All the benchmarks used in this paper are widely used for out-of-distribution generalization. Regarding baselines for finetuning CLIP,  I am not the best judge if all the relevant baselines have been used, since I am not familiar with recent works on CLIP finetuning.

**Other Comments Or Suggestions:**

- Given that the theoretical results (Theorem 5.3, 5.4) are mostly an application of existing theoretical results, I suggest the authors should rename the theorems to propositions.

- Just like the authors mention that Theorem 5.3 aligns with results in prior works, the same should should be done for Theorem 5.4 with the prior work by Ahuja et al. 2023 on interventional causal representation learning.

**Other Strengths And Weaknesses:**

**Strengths**

- The paper is well written, with details about the proposed method easy to follow and the empirical findings are clear and easy to follow.


**Weaknesses**

- The core ideas behind CLIP-ICM are not very original, the theoretical results in the paper mostly build upon existing proof techniques in the literature. Even the methodology of extract invariant features from representation with linear identification guarantees is not very novel. However, I don't think this is a major concern, as the application of identifiable invariant feature learning specifically to CLIP framework is novel to the best of my knowledge.

**Questions For Authors:**

- The description in section 4 about the domain shift is a bit confusing. The authors mention with linear probe the CLIP embeddings are kept frozen (line 153), but when analyzing the results they mention finetuning CLIP (line 168). Were the CLIP representations finetuned with linear probe or only a linear probe was trained with frozen representations?

- Table 1, open class domain shift scenario, why does finetuning improve the performance for base classes but it deteriorates the performance for novel classes?

**Relation To Broader Scientific Literature:**

The papers utilizes the methodology of invariant predictor learning, a fairly common approach for tackling out-of-distribution generalization.
Specifically, identifiable invariant predictor learning approaches have been proposed in prior works [1, 2]. The key contribution of the paper is to apply these ideas in the framework of CLIP.

References

- [1] Lu, Chaochao, Yuhuai Wu, José Miguel Hernández-Lobato, and Bernhard Schölkopf. "Invariant causal representation learning for out-of-distribution generalization." In International Conference on Learning Representations. 2021.

- [2] Yao, Dingling, Dario Rancati, Riccardo Cadei, Marco Fumero, and Francesco Locatello. "Unifying Causal Representation Learning with the Invariance Principle." arXiv preprint arXiv:2409.02772 (2024).

**Theoretical Claims:**

Yes, I checked the correctness of the proof for all the theorems and I did not find any major issues.

---

> ### Author Rebuttal · Authors · 2025-03-31
>
> We thank the reviewer for their thoughtful evaluation and positive feedback. We appreciate the acknowledgment that our approach offers a **solid** theoretical foundation and demonstrates **clear** empirical benefits for OOD generalization. We also value the reviewer’s recognition that our **claims are well-supported** by both theoretical analysis and practical experiments, as well as the confirmation that our references to existing literature provide **sufficient** context. Moreover, we are pleased that the reviewer finds our writing to be **clear** and our explanation of the proposed method to be **thorough**.
>
> Below, we address the reviewer’s additional questions and suggestions in detail.
>
> ## Response to Weaknesses
>
> We appreciate the reviewer’s positive comments and would like to clarify our position. We acknowledge that Theorem 5.3 and Theorem 5.5 indeed build upon previous work in the literature. However, our primary interest lies in extending these interesting theoretical insights to practical applications.
>
> In our manuscript, we carefully discuss the conditions under which Theorem 5.5 and Theorem 5.6 hold, and we leverage these conditions to propose our CLIP-ICM method, which is designed to guarantee lower OOD generalization error. One particularly surprising and encouraging observation is that by mapping both image and text embeddings into a shared invariant subspace, CLIP is able to maintain its original zero-shot performance even when confronted with domain shift—thus, ensuring that it continues to perform well on new classes after task-specific fine-tuning.
>
> We are grateful for the reviewer’s recognition of our work and believe that integrating theoretical results with real-world application strategies represents a significant contribution to the field.
>
> ## Response to Other Comments or Suggestion
>
> 1. > Given that the theoretical results (Theorem 5.3, 5.4) are mostly an application of existing theoretical results, I suggest the authors should rename the theorems to propositions.
>
> Thank you for your suggestion. In the final version, we will change these two theorems into propositions.
>
> 2. > the same should be done for Theorem 5.4 with the prior work by Ahuja et al. 2023 on interventional causal representation learning.
>
> We thank the reviewer for this suggestion. In the final version of the manuscript, we will explicitly highlight both the connections and distinctions between our Theorem 5.4 and the interventional causal representation learning work by Ahuja et al. (2023).
>
> ## Response to Questions For Authors
>
> 1. >  Were the CLIP representations finetuned with linear probe or only a linear probe was trained with frozen representations?
>
> We apologize for the confusion caused by our description, and thank the reviewer for highlighting this issue.
>
> To clarify, in the domain-shift scenario (line 153), the CLIP embeddings remain frozen, and only the linear probe is trainable. At line 168, when we mentioned "fine-tuning," we were referring specifically to the linear probe training process, rather than updating the original CLIP image encoder or text encoder.
>
> We will carefully revise this description in the final manuscript to clearly differentiate these two settings and avoid further confusion.
>
> 2. > Table 1, open class domain shift scenario, why does finetuning improve the performance for base classes but it deteriorates the performance for novel classes?
>
> We thank the reviewer for raising this insightful question. This phenomenon—where fine-tuning improves performance on the base classes but degrades performance on novel classes—has been widely acknowledged in studies on adapting CLIP, including CoOP, CoCoOp, CLIP-Adapter, and CLIPood. As noted by Wortsman et al. [1] and Shu et al. [2], naively fine-tuning CLIP often results in a loss of its inherent strong generalization ability, manifesting as improved performance on the specifically fine-tuned downstream task but significantly weakened robustness under distribution shift (including both covariate shift and label shift).
>
> A likely explanation for this deterioration on novel classes is tied to **catastrophic forgetting**, a phenomenon wherein a model “forgets” previously learned information when trained on new data. In the context of Table 1, when adapting CLIP to  a set of base classes, the fine-tuning procedure heavily optimizes for accurate classification of those base classes. Consequently, the original parameters—particularly those responsible for generalizing to unseen classes—are overwritten. As a result, the previously robust zero-shot capability of CLIP (which was central to its strong open-class performance) is compromised.
>
>
>
> [1] Wortsman, Mitchell, et al. Robust fine-tuning of zero-shot models. CVPR 2022.
>
> [2] Shu, Yang, et al. Clipood: Generalizing clip to out-of-distributions. ICML 2023.

---

> > ### Comment · Reviewer_HsFQ · 2025-04-03
> >
> > Thanks a lot for the rebuttal! I think the paper is very interesting, technically sound, and make a good use for proposing methodology inspired from the latent identification literature. Hence, I retrain my rating and vouch for acceptance.

---

> > > ### Author Response · Authors · 2025-04-03
> > >
> > > Thank you for your thoughtful review and kind recognition. We truly appreciate the time and effort you dedicated—it means a lot to us and encourages our continued work!

---

### Official Review · Reviewer_q6oc · 2025-03-13

**Overall Recommendation:** 3

**Summary:**

This work is motivated from the OOD generalization issue in CLIP, it addresses this problem via learning an invariant causal mechanism and proposes CLIP-ICM framework, which includes collecting interventional data, estimating a linear projection matrix, and predicting in the invariant subspace. The proposed CLIP-ICM shows improvement in OOD datasets.

## update after rebuttal
I appreciate the authors for their response, which addresses most of my concerns. I intend to maintain my original score.

**Claims And Evidence:**

In general, the claims are well supported. The paper originates from a well-studied principle in invariant learning, the high-level idea is not new, but it would still contribute to the CLIP model generalization.

**Essential References Not Discussed:**

N/A

**Experimental Designs Or Analyses:**

The experiment would be improved if further diverse environments and contexts are included.

**Methods And Evaluation Criteria:**

The pipeline of the method is generally clear, but some of the technical details may need further clarification. For example, it would be better to include a more detailed description of the interventional data generation process.

**Other Comments Or Suggestions:**

There are a few informal writing styles, e.g., the footnote in page 4 takes a single sentence in the main text.

**Other Strengths And Weaknesses:**

In general, the paper would be further improved with a more comprehensive evaluation.

**Questions For Authors:**

Could the authors share more details of the environment diversity and applicable scenarios of the proposed method?

**Relation To Broader Scientific Literature:**

The paper is related to CLIP applications in different domains.

**Theoretical Claims:**

The theoretical analysis looks good to me.

---

> ### Author Rebuttal · Authors · 2025-03-31
>
> We thank the reviewer for the thoughtful comments and positive feedback. We are pleased that the reviewer recognizes our work as **well-supported**, highlighting our **clear** pipeline and **sound** theoretical analysis. Below, we provide detailed responses addressing the specific concerns raised by the reviewer.
>
> ## Response to Methods and Evaluation Criteria
>
> We appreciate the reviewer’s comments regarding the interventional data generation process. We would like to clarify the following points:
>
> 1. In the original manuscript, lines 369–371 (left column) briefly outline the steps for collecting image-based interventional data, while lines 381–384 (left column) and line 330 (right column) briefly describe the process for collecting text-based interventional data. And we mention that the detailed collection process is provided in Appendix H.1.
> 3. In Appendix H.1, we present detailed description of the collection procedures for both image-based and text-based interventional data:
>    1. For the image-based interventional data, we explain that it is generated using eight data augmentation techniques, including: ColorJitter, GrayScale, GaussianBlur, RandomInvert, RandomRotation, RandomPosterize, RandomSolarize, and RandomEqualiz. These data augmentation techniques directly implemented from the *torchvision.transform* package.
>    2. The text-based interventional data comprises two components: the text description model and the text intervention model. Both models are generated by invoking GPT-4o. The prompts used for these models are provided on page 21, lines 1118–1142, and Figure 5 presents an example of the text-based interventional data.
>
> Please let us know if you have any additional suggestions for further improvements regarding this aspect.
>
> ## Response to Experimental Designs Or Analyses
>
> To address your concerns, in addition to the datasets used in the original manuscript (PACS, VLCS, OfficeHome, Terra Incognita, DomainNets, ImageNet, ImageNet-V2, ImageNet-S, ImageNet-A, and ImageNet-R), we additionally conducted experiments on the iWildCam-WILDS 2020 dataset.
>
> iWildCAM comprises 203,029 images from 182 different animal species, which were collected from 323 camera traps distributed across various locations. The images obtained from different locations exhibit variations in lighting, color, camera angle, background, vegetation, and relative animal frequencies.
>
> We follow the setting of Koh et al. (2021)[1], use images from 243 locations as the training domain and those from 48 other locations as the test domain. We report the average macro F1 score of CLIP, CLIP-ICM$^*$, CLIP-ICM$^\dagger$, and CLIP-ICM under both ID and OOD conditions, as shown in the table below:
>
> | Method| ID (48 Locations) | OOD (243 Locations) |
> | :- | :-: | :-: |
> | CLIP|14.2|10.6|
> | CLIP Linear-Probe|54.6|41.4|
> | CLIP-ICM$^*$|15.6|13.3|
> | CLIP-ICM$^*$ Linear-Probe|56.2|42.1|
> | CLIP-ICM$^\dagger$|15.2|12.2|
> | CLIP-ICM$^\dagger$ Linear-Probe|55.6|44.3|
> | CLIP-ICM|15.8|14.1|
> | CLIP-ICM Linear-Probe|57.1|46.1|
>
> [1] Koh et al. Wilds: A benchmark of in-the-wild distribution shifts. ICML 2021.
>
> ## Response to Other Strengths And Weaknesses
>
> Thank you for suggesting a more comprehensive evaluation. We have extensively validated our method across multiple datasets, including PACS, VLCS, OfficeHome, Terra Incognita, DomainNets, ImageNet, ImageNet-V2, ImageNet-S, ImageNet-A and ImageNet-R along with detailed ablation studies.
>
> Additionally, we have included experiments on the iWildCam-WILDS 2020 dataset in our previous response. Moreover, in our reply to Reviewer Pp7f, we added ablation studies concerning the role of $A_{inv}$. Please let us know if you have any further suggestions regarding other experiments.
>
> ## Response to Other Comments Or Suggestions
>
> We thank the reviewer for pointing out this issue. In the final version, we will check all formatting issues and make the necessary revisions.
>
> ## Response to Questions For Authors
>
> We thank the reviewer for the question and are happy to elaborate.
>
> 1. The environmental diversity across our evaluation datasets comes in different ways. For datasets like PACS, Office-Home, DomainNet, ImageNet-Sketch, and ImageNet-R, diversity primarily stems from variations in visual styles. In VLCS and Terra Incognita, it is reflected in background complexity, lighting conditions, and camera viewpoints. For ImageNet V2 and ImageNet-A, diversity arises from changes in image sources and the inclusion of hard-to-classify samples, respectively.
> 2. Our method is generally applicable to real-world scenarios where environment-induced distribution shifts occur. Potential applications include monitoring in wildlife habitats, perception systems in autonomous driving, and cross-domain image-text retrieval. In particular, tasks that require stable semantic understanding across diverse environments can benefit from the CLIP-ICM framework’s ability to isolate invariant semantic factors from CLIP representations.

---

### Official Review · Reviewer_Pp7f · 2025-03-16

**Overall Recommendation:** 3

**Summary:**

This paper introduces CLIP-ICM, a framework that improves CLIP’s OOD robustness by leveraging a causal perspective to separate invariant and variant factors. By learning a linear mapping to the invariant subspace using interventional data, CLIP-ICM enhances performance across multiple OOD datasets.

**Claims And Evidence:**

To the best of my knowledge, the evidence supports the claims well.

**Essential References Not Discussed:**

n/a

**Experimental Designs Or Analyses:**

I have checked section 7 and believe it follows the community standard

**Methods And Evaluation Criteria:**

To the best of my knowledge, the evaluation follows the community convention.

**Other Comments Or Suggestions:**

see above

**Other Strengths And Weaknesses:**

1. I am curious on the role of A_inv and the interventional data. After I check the ablation study, seems like there is no ablation study on this level. would be beneficial to present the ablation of the three steps in figure 3 to help reader to understand the importance of each component.

2. Where does the variance come from in the data reported in Table 2 and 3? Is it from the difference of interventional data or different initialization? It would be beneficial to have clarity don't that

2. Overall the paper's presentation is very clear and comprehensive, and studies an important problem that lies in the interest of the community.

**Questions For Authors:**

see above

**Relation To Broader Scientific Literature:**

The paper studies the OOD from a causal inference perspective, which bridges the gap between the two fields

**Theoretical Claims:**

I have checked section 5.1 and do not find any issue

---

> ### Author Rebuttal · Authors · 2025-03-31
>
> We thank the reviewer for the constructive feedback and valuable suggestions. We sincerely appreciate the reviewer for their positive feedback, especially for finding our claims **well-supported**, recognizing the **clarity** and **comprehensiveness** of our paper's presentation, affirming that our evaluation methodology **aligns with community standards**, and highlighting our contribution in **bridging** causal inference and OOD generalization. Additionally, we provide detailed responses to address the two specific concerns raised by the reviewer as follows.
>
> ## Response to Other Strengths And Weaknesses
>
> ### W1:
>
> We appreciate the reviewer's interest in understanding the contribution of the linear projection matrix $A_{inv}$ and the role of interventional data. Regarding the role of interventional data, we would like to first emphasize a few points:
>
> 1. Interventional data and the linear projection matrix $A_{inv}$ are  mutually dependent components of our method. According to Equation (9), without interventional data, it is unlikely to estimate $A_{inv}$ in our framework.
> 2. As shown in Tables 2, 3, 5, 6, and 10–14, we have conducted extensive comparisons between three variants of our method utilizing different type of interventional data. Specifically:
>    - CLIP-ICM$^*$: using only image-based interventional data,
>    - CLIP-ICM$^\dagger$: using only text-based interventional data,
>    - CLIP-ICM: using both types of interventional data.
> 3. We have provided an ablation study on the effect of different numbers of interventional data pairs in Appendix M, Figure 6 (d).
>
> Regarding the role of $ A_{inv} $, we agree with your suggestion and thus we include an additional ablation experiment to further illustrate its importance. Specifically, we use our generated image-based interventional data to train a linear-probe on the DomainBed dataset.  The experimental results are summarized as follows.
>
> | Method                              |   PACS   |   VLCS   | OfficeHome | TerraInc | DomainNet |   AVG.   |
> | :---------------------------------- | :------: | :------: | :--------: | :------: | :-------: | :------: |
> | Linear Probe                        |   96.4   |   78.7   |    81.9    |   60.2   |   55.0    |   74.4   |
> | Linear Probe + Interventional  data |   96.8   |   79.3   |    82.3    |   60.5   |   55.8    |   74.9   |
> | CLIP-ICM$^*$ + Linear Probe         | **97.5** | **86.5** |  **84.6**  | **64.3** | **64.0**  | **79.0** |
>
> From the results, we can observe that:
>
> 1. Incorporate image-based interventional data with linear probe only slightly (0.5%) improve the performance of linear probe.
> 2. Despite all incorporate image-based interventional data for training, the performance of CLIP-ICM$^*$ + Linear Probe is significantly better than that of Linear Probe + Interventional data.
>
> These findings demonstrate that our proposed $A_{inv}$ module (i.e., the projection to the invariant subspace) indeed improve the performance of CLIP in OOD scenarios.
>
> ### W2:
>
> Regarding the source of variance in Tables 2 and 3, each value in Tables 2 and 3 represents the mean and standard deviation over 5 runs with different random seeds. According to the standard evaluation protocol of the DomainBed Benchmark, we believe that the primary source of variance originates from the various splits of the training, validation, and test datasets across multiple runs.

---

### Decision · Program_Chairs · 2025-05-01

**Decision:**

Accept (poster)

**Comment:**

This paper proposes a causal mechanism for addressing CLIP's out-of-distribution (OOD) problem. The paper is well written, and reviewers noted the strong empirical evidence demonstrating the effectiveness of the proposed method compared to naive finetuning strategies and other baseline models. However, the theoretical claims were not unanimously accepted by reviewers. In particular, one reviewer raised significant criticisms regarding the theoretical foundation of the work, its assumptions, and the proposed theorems. During discussions with reviewers, it became clear that while these theoretical uncertainties are important, they may not undermine the strength of the empirical results. Moreover, reviewers disagreed among themselves about the theoretical merits of the work. Given these considerations, I believe the paper represents an interesting contribution to the field and therefore recommend acceptance. However, the authors should carefully address the theoretical concerns raised by reviewers in their revised manuscript.